# ADARL: ADAPTIVE LOW-RANK STRUCTURES FOR ROBUST POLICY LEARNING UNDER UNCERTAINTY

## ABSTRACT

Robust reinforcement learning (Robust RL) seeks to handle epistemic uncertainty in environment dynamics, but existing approaches often rely on nested min–max optimization, which is computationally expensive and yields overly conservative policies. We propose **Adaptive Rank Representation (AdaRL)**, a bi-level optimization framework that improves robustness by aligning policy complexity with the intrinsic dimension of the task. At the lower level, AdaRL performs policy optimization under fixed-rank constraints with dynamics sampled from a Wasserstein ball around a centroid model. At the upper level, it adaptively adjusts the rank to balance the bias–variance trade-off, projecting policy parameters onto a low-rank manifold. This design avoids solving adversarial worst-case dynamics while ensuring robustness without over-parameterization. Empirical results on MuJoCo continuous control benchmarks demonstrate that AdaRL not only consistently outperforms fixed-rank baselines (e.g., SAC) and state-of-the-art robust RL methods (e.g., RNAC, Parseval), but also converges toward the intrinsic rank of the underlying tasks. These results highlight that adaptive low-rank policy representations provide an efficient and principled alternative for robust RL under model uncertainty.

## 1 INTRODUCTION

The goal of a reinforcement learning (RL) agent is to learn a policy that maximizes its expected discounted cumulative reward (Sutton et al., 1998). Recent advances have enabled RL agents to master complex games and robotic control tasks in both simulation and the real world (Mnih et al., 2015; Silver et al., 2017). However, policies that perform well in such controlled settings often fail to transfer to practice, where transition dynamics are rarely fixed and may shift due to modeling inaccuracies (Lanzani, 2025), external disturbances, or changing conditions (Pattanaik et al., 2017). To address this gap, robust reinforcement learning (robust RL) (Zhou et al., 1996) formalizes uncertainty by considering a set of possible transition kernels and casting policy optimization as a minmax problem: the agent seeks a policy that maximizes expected return under the worst-case dynamics. This formulation reduces the sensitivity of RL to model misspecification and aims to produce policies that stay reliable when the environment differs from training.

Robust RL provide a principled framework to handle model uncertainty by optimizing for policies that perform well under the worst-case transition models within a prescribed uncertainty set (Iyengar, 2005; Wiesemann et al., 2013). Classical solutions extend Bellman's principle to robust settings (Satia & Lave Jr, 1973), while more recent work has focused on robust policy learning via model-based planning (Clavier et al., 2023) or online interaction with a nominal environment (Wang & Zou, 2021). Despite these advances, robust RL faces severe scalability issues when applied to continuous and high-dimensional domains. In particular, updating the robust value function via the robust Bellman operator requires solving a nested inner-loop optimization at every step, i.e., identifying the worst-case transition, which becomes computationally prohibitive as the state and action spaces grow or when the uncertainty set is large or unbounded (Wang & Zou, 2022). Moreover, existing approaches often assume access to oracle solvers or rely on fixed uncertainty sets that may yield overly conservative policies (Mannor et al., 2012; 2016; Xu & Mannor, 2012). Beyond these computational bottlenecks, another key challenge lies in function approximation. Existing analyzes are mostly restricted to the tabular setting, which cannot achieve parameterized neural network approximations to the optimal solution of the robust Bellman equation. Our approach,

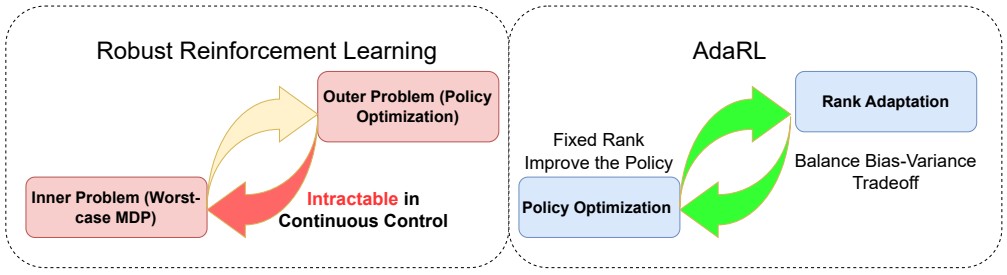

Figure 1: Comparison between classical robust reinforcement learning and the proposed AdaRL framework. Robust RL requires solving a nested min–max optimization, where the inner worst-case MDP search becomes intractable in continuous control. In contrast, AdaRL replaces this inner optimization with an adaptive low-rank mechanism and alternates between policy optimization and rank adaptation to achieve a scalable and data-driven robustness under epistemic uncertainty.

in contrast, explicitly accommodates parameterization, thereby enabling robust generalization in high-dimensional environments.

In this work, we introduce an alternative perspective to overcome the limitations of existing robust RL approaches. Instead of directly tackling worst-case dynamics through nested min–max optimization, we enhance robustness by controlling over-parameterization and improving the generalization of fixed-rank policy and value models under perturbed transition dynamics. A key insight (Li et al., 2018)) is that the effective complexity of a policy should *match* the intrinsic dimension of the task under epistemic uncertainty—uncertainty in environment dynamics arising from limited data or partial observability, which is prevalent in real-world domains such as robotics, control, environmental policy, and economics (Nagami & Schwager, 2023; Zhou et al., 1996; Lemoine & Traeger, 2014; Hansen & Sargent, 2008). Building on this idea, we propose a new algorithm that jointly learns both the policy models and its rank, formulated as a bi-level optimization problem: the lower-level learns a policy under low-rank constraints, while the upper-level adapts the rank to balance robustness and expressiveness.

This perspective aligns with and extends prior work on exploiting low-rank structures in reinforcement learning. In the *model-based* setting, algorithms for joint feature and policy learning have been developed when the dynamics admit a low-rank decomposition (Agarwal et al., 2020; Bose et al., 2024). In the *model-free* setting, Jiang et al. (2017) introduced the concept of *Bellman rank* to capture the intrinsic complexity of value function approximation, and subsequent work (Modi et al., 2021; 2024; Yang et al., 2020) sought to encourage small Bellman rank during training. More recently, Tiwari et al. (2025) showed that wide two-layer neural networks yield reachable states concentrated on a low-dimensional manifold whose dimension scales with the action space. Overall, these works show that low-rank structures can improve performance in *standard RL settings*. Yet, no existing approach provides a practical algorithm for leveraging low-rank advantages under *model uncertainty*, and it remains inherently difficult to determine a suitable rank for parameterizing policy models in uncertain environments.

**Our Contribution.** We propose **Adaptive Rank Representation for Reinforcement Learning** (AdaRL, Figure 1), an adaptive framework that integrates conservatism into the learning process in MDPs with epistemic uncertainty. The algorithm alternates between standard policy optimization under a fixed rank and an adaptive step that adjusts the rank to balance robustness and expressiveness. Our main contributions are:

1. We provide a theoretical analysis of the bias–variance trade-off in entropy-regularized RL with linear parameterization under epistemic uncertainty, showing that low-rank representations can reduce variance in the presence of model uncertainty (Section 3, Theorem. 1).

2. We formulate policy rank selection as a bi-level optimization problem and present the AdaRL algorithm, which adaptively adjusts policy rank for robust learning (Section 4).

3. We empirically evaluate AdaRL on standard MuJoCo continuous control benchmarks, demonstrating consistent improvements over robust baselines (e.g., RNAC Zhou et al. (2023), Parseval Chung et al. (2024)) and non-robust methods such as SAC (Haarnoja et al., 2018) and Tiwari et al. (2025) (see Section 5).

## 2 PRELIMINARY AND RELATED WORKS

### 2.1 NOTATION

**Discounted Markov Decision Process.** A Markov decision process (MDP) is represented by the tuple $(\mathcal{S}, \mathcal{A}, P, \rho, r, \gamma)$ wherein $\mathcal{S}$ is the state space $\mathcal{A}$ is the action space (with both $\mathcal{S} \subset \mathbb{R}^n, \mathcal{A} \subset \mathbb{R}^m$ assumed compact), $P_{s,a} \in \Delta_{\mathcal{S}}$, is the transition kernel for $a \in \mathcal{A}, s \in \mathcal{S}$. (where $\Delta_{\mathcal{S}}$ denotes the space of probability measures with support $\mathcal{S}$), $\rho(\cdot)$ is the initial state distribution, $R : \mathcal{S} \times \mathcal{A} \to \mathbb{R}$ is the the reward function $\gamma \in [0, 1)$ is the discount factor. Given $s \in \mathcal{S}$, a policy $\pi$ is a map $\pi(\cdot|s) : \mathcal{S} \to \Delta_{\mathcal{A}}$, where $\Delta_{\mathcal{A}}$ denotes the space of probability measures with support $\mathcal{A}$.

**Epistemic Uncertainty in State Dynamics.** To model uncertainty in the environment dynamics, we introduce an ambiguity set of possible transition kernels:

$$\mathcal{P}_{s,a} := \{P_{s,a} \in \Delta_{\mathcal{S}} \mid W(\hat{P}^{\circ}_{s,a}, P_{s,a}) \le \epsilon\},$$

where $\hat{P}^{\circ}_{s,a}$ is a reference transition kernel (e.g., a maximum likelihood estimator obtained from a finite demonstration dataset), $W(\hat{P}^{\circ}_{s,a}, P_{s,a})$ denotes the Wasserstein distance (Villani et al., 2008), and $\epsilon > 0$ is the uncertainty radius. We refer to $P^{\circ}_{s,a}$ as the centroid of the uncertainty set, representing the true but unobserved transition kernel that governs the system dynamics. Throughout, we assume that epistemic uncertainty is well captured by the Wasserstein ball (Mohajerin Esfahani & Kuhn, 2018), i.e., $P^{\circ}_{s,a} \in \mathcal{P}_{s,a}$ for all $(s, a) \in \mathcal{S} \times \mathcal{A}$.

**Singular Value Decomposition (SVD)** Let $\theta \in \mathbb{R}^{d_1 \times d_2}$. A *thin* singular value decomposition (SVD) is given by $\theta = \mathbf{U}\mathbf{\Sigma}\mathbf{V}^{\top}$, where $\mathbf{U}$ is a $d_1 \times r$ matrix with orthogonal columns, that is, an element of the Stiefel manifold (Chakraborty & Vemuri, 2019; Atiyah & Todd, 1960)

$$\mathrm{St}(r, d_1) = \{\mathbf{U} \in \mathbb{R}^{d_1 \times r} : \mathbf{U}^T\mathbf{U} = \mathbf{I}\},$$

$\mathbf{\Sigma}$ is a $r \times r$ diagonal matrix with positive entries $\sigma_1 \ge \sigma_2 \ge \cdots \sigma_r > 0$ (referred to as singular values) and $\mathbf{V} \in \mathrm{St}(r, d_2)$. The singular value decomposition exists for any matrix $\theta \in \mathbb{R}^{d_1 \times d_2}$. We refer to a *truncated* SVD whenever $r < \mathrm{rank}(\theta)$.

### 2.2 ROBUST REINFORCEMENT LEARNING

In MDPs, the system dynamics $P$ is usually assumed to be constant over time. However, in the real world, it is subject to perturbations that can significantly impact performance in deployment (Zhang et al., 2023; Moos et al., 2022). Robust MDPs provide a theoretical framework for taking this uncertainty into account, taking $P$ as not fixed but chosen adversarially from an uncertainty set $\mathcal{P}$ (Iyengar, 2005; Nilim & El Ghaoui, 2005), where $\mathcal{P}$ denotes a set of plausible transition models known as the uncertainty set. The objective of robust RL is to find a policy that performs well under the worst-case dynamics within this set. Formally, the robust objective $\mathcal{J}_{\mathcal{P},\pi}$ is defined as:

$$\mathcal{J}_{\mathrm{robust}}(\pi) = \max_{\pi} \min_{P \in \mathcal{P}} \mathbb{E}_{P,\pi}\Big[ \sum_{t \ge 0} \gamma^t R(s_t, a_t) \Big| s_0 \sim \rho_0 \Big] \tag{1}$$

The optimal policy $\pi^*_{\mathcal{P}}$ is defined as the solution to the *outer-loop* problem, which maximizes $\mathcal{J}_{\mathrm{robust}}(\pi)$ by accounting for the worst-case transition model at each time step. This leads to the *inner-loop* problem of identifying the worst-case dynamics, for which several approaches have been developed, including value iteration (Nilim & El Ghaoui, 2005; Iyengar, 2005; Wiesemann et al., 2013; Grand-Clément & Kroer, 2021; Kumar et al., 2023a), policy iteration (Kumar et al., 2022; Badrinath & Kalathil, 2021), and policy gradient methods (Li et al., 2022; Wang & Zou, 2022; Wang et al., 2023; Kumar et al., 2023b). However, the problem remains NP-hard for general uncertainty sets, and optimal policies may even be non-stationary (Wiesemann et al., 2013). Most existing methods sidestep this difficulty by assuming that the inner-loop optimization can be solved efficiently—a reasonable assumption in tabular settings with small uncertainty sets, where one can exhaustively evaluate all transition kernels $P \in \mathcal{P}$. Yet, when the uncertainty set is continuous, the inner-loop problem becomes substantially more challenging and computationally expensive. To address this challenge, Zhou et al. (2023); Gadot et al. (2024) propose the RNAC and EWoK algorithms, which rely on sampling-based techniques to estimate value functions under worst-case dynamics. Although theoretically sound, these methods require drawing multiple next states for each state-action pair, leading to high sample complexity and considerable computational overhead.

## 2.3 Reinforcement Learning with low rank structure

Another direction of research to address this uncertainty is to take advantage of *low-rank structures in dynamics*. In many stochastic control tasks, the transition dynamics admit a low-rank decomposition over a finite set of state-action features (Rozada et al., 2024; 2021; Yang et al., 2019). For example, Tiwari et al. (2025) show that under suitable assumptions, the set of attainable states lies on a low-dimensional manifold. In fixed environments, the dimension of this manifold grows only linearly with the size of the action space and is independent of the state-space dimension. Building on this observation, they employ a $(2d_a + 1)$-dimensional low-rank manifold and apply sparse reinforcement learning methods to solve MuJoCo control tasks. More generally, low-rank structure can be imposed either on the transition kernel or directly on the optimal action-value function $Q^*$, and empirical evidence suggests that $Q^*$ and near-optimal Q-functions in common stochastic control tasks indeed exhibit low-rank properties (Sam et al., 2023; Rozada et al., 2024; 2021; Yang et al., 2019).

Motivated by these findings, algorithms for joint feature and policy learning in *model-based* RL have been developed (Agarwal et al., 2020; Bose et al., 2024), though they typically assume the rank is known a priori. For *model-free* RL, Jiang et al. (2017) introduced the notion of *Bellman rank* to quantify the intrinsic complexity of value function approximation. More recent approaches exploit low-rank factorizations or representations to implicitly encourage small Bellman rank while optimizing the policy or value function (Modi et al., 2021; 2024; Yang et al., 2020). However, the theoretical guarantees in these works generally rely on fixed dynamics, and to date there is no algorithm that simultaneously recovers the exact Bellman rank while learning the optimal policy under uncertain or time-varying environments.

## 3 Bias-Variance Tradeoff in RL with Epistemic Uncertainty

As highlighted in the related work section, many control tasks naturally admit low-rank structures in their transition dynamics, which has motivated a line of methods leveraging fixed-rank representations. However, when moving to the robust MDP setting, the presence of epistemic uncertainty fundamentally changes the picture. On the one hand, adopting an excessively low rank may fail to capture the variability introduced by uncertain dynamics, leading to biased estimates and brittle policies. On the other hand, employing a large rank increases model expressiveness but also amplifies variance, making the policy highly sensitive to perturbations and prone to over-parameterization. This tension suggests that selecting an appropriate rank is crucial: the rank must be sufficiently rich to encode uncertainty, yet controlled enough to mitigate overfitting. In this section, we formally analyze this bias–variance tradeoff in reinforcement learning under epistemic uncertainty, beginning with the model-free setting of entropy-regularized reinforcement learning (Haarnoja et al., 2018). The objective function of entropy-regularized reinforcement learning is given by:

$$J(\pi) = \mathbb{E}_\pi \left[ \sum_{t=0}^{\infty} \gamma^t \Big( R(s_t, a_t) + \mathcal{H}(\pi(\cdot|s_t)) \Big) \right], \tag{2}$$

For any given policy $\pi$, we define the corresponding (entropy regularized) $Q^\pi$ function and $V^\pi$ function as follows:

$$V^\pi(s) = \mathbb{E}_{a_t \sim \pi(\cdot|s_t), s_{t+1} \sim \mathcal{P}_{s_t, a_t}} \left[ \sum_{t \geq 0} \gamma^t \Big( R(s_t, a_t) + \mathcal{H}(\pi(\cdot|s_t)) \Big) \Big| s_0 = s \right] \tag{3}$$

$$Q^\pi(s, a) = \mathbb{E}_{a_t \sim \pi(\cdot|s_t), s_{t+1} \sim \mathcal{P}_{s_t, a_t}} \left[ \sum_{t \geq 0} \gamma^t \Big( R(s_t, a_t) + \mathcal{H}(\pi(\cdot|s_t)) \Big) \Big| s_0 = s, a_0 = a \right] \tag{4}$$

where we write $s_{t+1} \sim \mathcal{P}_{s_t, a_t}$ to indicate that a transition kernel $P_{s_t, a_t}$ is uniformly randomly sampled from the uncertainty set $\mathcal{P}_{s_t, a_t}$ and $s_{t+1} \sim P_{s_t, a_t}$ and the entropy term is defined as $\mathcal{H}(\pi(\cdot|s_t)) := -\sum_{a \in \mathcal{A}} \pi(a|s_t) \log \pi(a|s_t)$. Let $\pi^*$ denote the optimal policy. We begin by restating a well known characterization of the solution to the entropy regularized MDP. According to Haarnoja et al. (2018), the optimal policy takes the following form:

$$\pi^*(a|s) = \exp \big( Q^*(s, a) - V^*(s) \big) \tag{5}$$

where $Q^*$ is the unique fixed point of the *soft* Bellman operator

$$\mathcal{B}Q(s, a) := R(s, a) + \gamma \mathbb{E}_{s' \sim \mathcal{P}_{s, a}} \left[ \log \sum_{a' \in \mathcal{A}} \exp Q(s', a') \right] \tag{6}$$

and $V^*(s') := \log \sum_{a' \in \mathcal{A}} \exp Q(s', a')$. We consider linear function approximations for $Q(s, a)$ and $V(s)$ functions for the simplicity of analysis, i.e.:

$$Q_\theta(s, a) = \phi(s, a)^\top \theta \quad \text{and} \quad V_\omega(s) = \psi(s)^\top \omega$$

where $\phi(s, a)$ and $\psi(s)$ are feature mappings.

**Assumption 1** *We assume the training data in the form of triplets $(s, a, s')$ is generated as follows: $a \sim \pi_b(\cdot \mid s) > 0$ where $\pi_b$ is a behavioral policy and $s' \sim \mathcal{P}_{s,a}$. We assume the induced Markov chain is ergodic and the steady-state distribution of triplets $(s, a, s')$ is denoted by $\mathcal{P}$. Similarly, we denote by $\mathcal{P}^\circ$ the steady-state distribution of $(s, a, s')$ when $a \sim \pi_b(\cdot \mid s), s' \sim P^\circ_{s,a}$.*

Hence, in an off-policy setting, the optimal policy with linear function approximation can be described as the solution to the following optimization problem:

$$\min_\omega \quad \mathbb{E}_\mathcal{P}\Big[\|\psi(s')^\top \omega - \log \sum_{a' \in \mathcal{A}} \exp \phi(s', a')^\top \theta^*(\omega)\|^2\Big] \tag{7}$$

$$\text{s.t} \quad \theta^*(\omega) = \arg\min_\theta \mathbb{E}_\mathcal{P}\Big[\|R(s, a) + \gamma\psi(s')^\top \omega - \phi(s, a)^\top \theta\|^2\Big] \tag{8}$$

where $\mathcal{P}$ denotes the steady-state distribution over $(s, a, s')$ induced by uniformly sampling transition kernels from the Wasserstein ball and executing a fixed behavioral policy. For simplicity, we write $\mathbb{E}_\mathcal{P} := \mathbb{E}_{(s,a,s',a') \sim \mathcal{P}, \pi_b}$. The first-order (sufficient) conditions for lower-level optimality can then be written as:

$$-\mathbb{E}_\mathcal{P}\big[\phi(s, a)(R(s, a) + \gamma\psi(s')^\top \omega - \phi(s, a)^\top \theta)\big] = 0 \tag{9}$$

wherein we write $\mathbb{E}_\mathcal{P}$ as shorthand for $\mathbb{E}_{(s,a,s') \in \mathcal{P}}$. This system of equations can be re-written as $A_\mathcal{P}\theta = b_{\mathcal{P},\omega}$ where

$$A_\mathcal{P} := \mathbb{E}_\mathcal{P}[\phi(s, a)\phi(s, a)^\top] \quad b_{\mathcal{P},\omega} := \mathbb{E}_\mathcal{P}[\phi(s, a)\big(R(s, a) + \gamma\psi(s')^\top \omega\big)]$$

Similarly for the ground-truth kernel $\mathcal{P}^\circ$ we define the system:

$$A_{\mathcal{P}^\circ} := \mathbb{E}_{\mathcal{P}^\circ}[\phi(s, a)\phi(s, a)^\top] \quad b_{\mathcal{P}^\circ,\omega} := \mathbb{E}_{\mathcal{P}^\circ}[\phi(s, a)\big(R(s, a) + \gamma\psi(s')^\top \omega\big)]$$

Our analysis investigates the consequences of using high-rank parametrized policies when the underlying ground-truth environment dynamics are of lower rank. Let $(\theta^\circ, \omega^\circ)$ denote the solution of the optimization problem defined by Eq. 7 and Eq. 8 when the expectations are taken with ground-truth dynamics. Let $(\theta_\mathcal{P}, \omega_\mathcal{P})$ denote the solution of the optimization problem defined by Eq. 7 and Eq. 8 when the expectations are taken with uniformly random sample from the Wasserstein ball centered at the reference Markov kernel $\hat{P}^\circ$ with radius $\epsilon$. To formalize this setting, we characterize the low-rank structure of the environment dynamics under a set of regularity conditions. In particular, we assume bounded feature mappings, nonsingular covariance matrices, and a discrete Picard condition, which are standard in reinforcement learning with linear function approximation.

**Assumption 2** *(2.1) $\|\phi(s, a)\| \leq 1, \forall(s, a) \in \mathcal{S} \times \mathcal{A}$.*
*(2.2) The feature covariance matrices with respect to ground truth dynamics are non-singular:*

$$\mathbb{E}_{\mathcal{P}^\circ}[\phi(s, a)\phi(s, a)^\top] \succ 0$$

*(2.3) (Lipschitz) $\forall(s, a) \in \mathcal{S} \times \mathcal{A}$, it holds that:*

$$\big\|\phi(s, a)^\top \theta_1 - \phi(s, a)^\top \theta_2\big\| \leq L\big\|\theta_1 - \theta_2\big\| \tag{10a}$$

*where $L > 0$. These are standard assumptions in reinforcement learning with linear function approximation Tsitsiklis & Van Roy (1996); Munos (2003).*

**Assumption 3** *(Discrete Picard Condition) The linear system $A_{\mathcal{P}^\circ}\theta = b_{\mathcal{P}^\circ,\omega^\circ}$ with $r^\circ := \text{rank}(A_{\mathcal{P}^\circ})$ satisfies the discrete Picard condition, i.e. the SVD $A_{\mathcal{P}^\circ} = U^\circ \Sigma_\mathcal{P} V^{\circ\top}$ is such that there exists $p > 1$ with:*

$$|u_i^{\circ\top} b_{\mathcal{P}^\circ,\omega^\circ}| \leq \sigma_{\mathcal{P}^\circ,i}^p \quad \text{for } i = 1, \dots, r^\circ,$$

$$|u^\circ_i{}^\top b_{\mathcal{P}^\circ,\omega^\circ}| \leq \sigma_{\mathcal{P}^\circ,r^\circ}^p \quad \text{for } i = r^\circ + 1, \dots, d.$$

The discrete Picard condition (Hansen, 1990; Levin & Meltzer, 2017) states that the magnitude of the inner product $|u^{\circ\top}_i b_{\mathcal{P}^\circ,\omega^\circ}|$ shrinks faster that $\sigma^p_i$, accounting for the ill-condition in the system dynamics. Here $p > 1$ describes the shrinking speed.

Building on these assumptions, we next examine the effect of approximating the system $A_\mathcal{P}\theta = b_\mathcal{P}$ using an $r$-truncated SVD decomposition of $A_\mathcal{P}$, denoted $A_{\mathcal{P},r}$. This result highlights the fundamental bias–variance trade-off: choosing too small an $r$ induces approximation bias, whereas choosing too large an $r$ amplifies estimation variance.

**Theorem 1** *Bias-Variance Trade-off of Rank-r Approximation:* *Assume the ground-truth dynamics are given by $\mathcal{P}^\circ$ and Assumptions 2 holds. Consider a truncated SVD $A_{\mathcal{P},r} = U\Sigma_{\mathcal{P},r}V^\top$ for $r \le \mathrm{rank}(A_\mathcal{P})$ and $\theta_r$ be the solution $A_{\mathcal{P},r}\theta = b_{\mathcal{P},\omega_\mathcal{P}}$. It holds that:*

$$\|\theta_r - \theta^\circ\|_2 \le \underbrace{\frac{1}{\sigma_{\mathcal{P},r}}\|(b_{\mathcal{P},\omega_\mathcal{P}} - b_{\mathcal{P}^\circ,\omega^\circ})\|_2}_{variance} + \underbrace{\|\sum_{i=r+1}^{d} v_i v_i^T \theta^\circ\|_2}_{bias} + 2\mathcal{O}(L\epsilon) \tag{11}$$

*If in addition Assumption 3 holds then:*

$$\|\theta_r - \theta^\circ\|_2 \le \underbrace{\frac{1}{\sigma_{\mathcal{P},r}}\|b_{\mathcal{P},\omega_\mathcal{P}} - b_{\mathcal{P}^\circ,\omega^\circ}\|_2}_{variance} + \underbrace{(d-r)\sigma^{p-1}_{\mathcal{P}^\circ,r} + (d-r)r^\circ\sigma^{p-1}_{\mathcal{P}^\circ,1}}_{bias} + 2\mathcal{O}(L\epsilon) \tag{12}$$

where $r^\circ := \mathrm{rank}(A_{\mathcal{P}^\circ})$, and $\epsilon > 0$ denotes the radius of the Wasserstein ball (Mohajerin Esfahani & Kuhn, 2018).

**Remark** The upper bound of the performance gap between the estimated parameter $\theta_r$ and the optimal solution $\theta^\circ$ in Theorem.1 can be decomposed into two components related to variance and bias respectively. Thus for example, the choice of $r > r^\circ$ introduces *higher* variance since $\sigma_{\mathcal{P},r} < \sigma_{\mathcal{P},r^\circ}$. Conversely, the choice of $r < r^\circ$ introduces *higher* bias since

$$(d-r)\sigma^{p-1}_{\mathcal{P}^\circ,r} + (d-r)r^\circ\sigma^{p-1}_{\mathcal{P}^\circ,1} > (d-r^\circ)\sigma^{p-1}_{\mathcal{P}^\circ,r^\circ} + (d-r^\circ)r^\circ\sigma^{p-1}_{\mathcal{P}^\circ,1}$$

**Discussion** To confirm that bias-variance tradeoff also exists in settings with non-linear representation, we perform a sanity check on a MuJoCo control task (Todorov et al., 2012). Specifically, we employ a three-layer neural network and adopt a rank-control mechanism similar to (Hu et al., 2022; Xu et al., 2019) (see details in Sec.4.2). Our experiments reveal a clear bias–variance tradeoff in nonlinear control models, as illustrated in Figure 2: models with extremely low-rank representations exhibit high bias, while high-rank models suffer from large approximation errors due to transition samples drawn from uncertain dynamics.

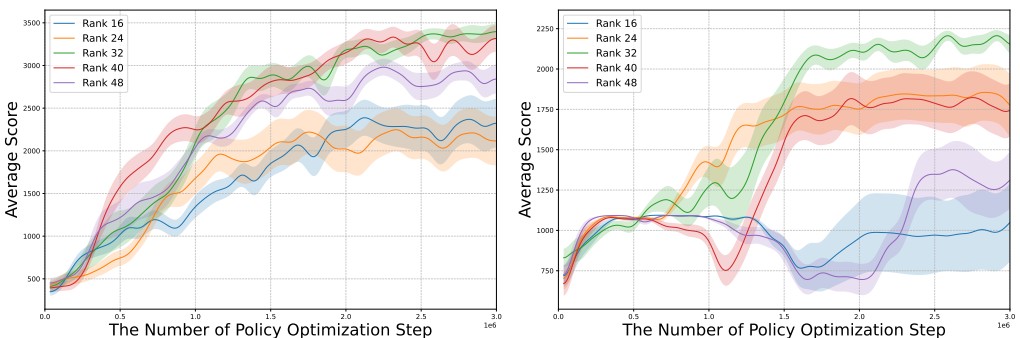

Figure 2: Performance of policy models under high model uncertainty in Walker2d-v3 (**Left**) and Hopper-v3 (**Right**). Results indicate that extremely low-rank representations lead to high bias, while overly high-rank models incur large approximation errors due to transition samples drawn from uncertain dynamics.

# 4 ADAPTIVE RANK REPRESENTATION REINFORCEMENT LEARNING

## 4.1 A BI-LEVEL OPTIMIZATION FORMULATION

The analysis in previous Section highlights that selecting the policy rank involves a delicate balance: too small a rank induces bias, while too large a rank amplifies variance. This trade-off suggests the need for an adaptive mechanism that can automatically adjust the rank during learning. Motivated by this insight, we introduce a bi-level (Colson et al., 2007) optimization formulation, where the lower-level problem identifies the optimal policy with uniformly sampled environment dynamics (from a Wasserstein ball around a centroid model) under a fixed rank, and the upper-level problem searches for the representation that optimizes a measure of fit to the lower-level model while regularizing by rank. To begin with, We consider a parameterized policy $\pi_\eta$, where $\eta \in \mathbb{R}^{d_1 \times d_2}$ with $d_1, d_2 > 0$. And we respectively denote by

$$\mathcal{M}_r := \{\eta \in \mathbb{R}^{d_1 \times d_2} \mid \text{rank}(\eta) = r\} \quad \mathcal{M}_{\leq \bar{r}} := \{\eta \in \mathbb{R}^{d_1 \times d_2} \mid \text{rank}(\eta) \leq \bar{r}\}$$

the smooth manifold of matrices with rank $r$ and the algebraic variety of matrices with rank less than or equal to $\bar{r} > 0$.

**Formulation:** Towards developing an approach that simultaneously learns the policy and adaptively adjusts its rank, we introduce the following bi-level formulation:

$$\min_r \mathbb{E}_{(s,a) \sim \mathcal{P}_{\eta^*}} \|\text{Proj}_{\mathcal{M}_r}(\pi_{\eta^*})(a|s) - \pi_{\eta^*}(a|s)\|_2 + \lambda r \tag{13}$$

$$\text{s.t.} \ \eta^* := \arg \max_{\eta \in \mathcal{M}_{\leq \bar{r}}} \mathbb{E}_{\tau \sim \mathcal{P}_{\pi_\eta}} \Big[ \sum_{t \geq 0} \gamma^t \Big( R(s_t, a_t) + \mathcal{H}(\pi_\eta(\cdot|s_t)) \Big) \Big] \tag{14}$$

where $\mathcal{P}_{\eta^*}$ denotes the steady-state distribution obtained by uniformly sampling the transition kernel from the Wasserstein ball and selecting actions according to the policy $\pi_{\eta^*}$, the operator $\text{Proj}_{\mathcal{M}_r}(\pi_{\eta^*})$ denotes the projection of the policy onto the low-rank manifold $\mathcal{M}_r$, $\lambda$ serves as a weight for rank regularization $r$, where $r$ denotes the rank variable, and $\bar{r}$ represents its maximum allowable value.

**Discussion** The bi-level formulation in Eq. 13–Eq. 14 plays two complementary roles. The *lower-level problem* Eq. 14 optimizes the policy parameters under a fixed rank constraint, aiming to maximize the entropy-regularized return and thus capture the best achievable policy representation at that rank. However, the optimal solution $\pi_{\eta^*}$ of the lower-level problem may not align with the intrinsic task complexity and can overfit by exploiting the full representation power. To address this, the *upper-level problem* Eq. 13 explicitly searches for an appropriate rank that balances bias and variance, as motivated in the previous section. It seeks the best low-dimensional representation (bounded by $\bar{r} > 0$) of the state–action value associated with $\pi_{\eta^*}$, while controlling model capacity through the rank regularization term. In this way, the upper-level problem enforces a bias–variance tradeoff, ensuring that the learned representation achieves robustness without unnecessary over-parameterization.

## 4.2 ALGORITHM

We are now ready to design algorithms for the proposed formulation. Note that our formulation has a hierarchical structure and falls into the class of bi-level optimization problems Hong et al. (2023); Colson et al. (2007). In general, bi-level problems are challenging to solve; in our case, the upper-level objective Eq. 13 depends explicitly on the optimal solution of the lower-level problem. Furthermore, the rank regularizer $C(\cdot)$ is non-differentiable, which precludes the use of (stochastic) first-order methods for the upper-level optimization. Fortunately, as we will show, a simple yet effective adaptive greedy search algorithm can be employed to obtain an empirical solution to the upper-level problem. At a high level, the proposed algorithm alternates between two steps: a **Rank Adaptation Step**, which updates the rank $r$ via a greedy search procedure, and a **Policy Optimization Step**, which optimizes the parameters under the rank constraint $\eta \in \mathcal{M}_{\leq r}$. We now examine each step in detail.

**Rank Adaptation Step** From the discussion in Section 3, we know that extremely low-rank models are limited in their representation power and thus fail to capture sufficient information under model uncertainty. In contrast, high-rank models tend to overfit, resulting in poor generalization. Hence, it is crucial to carefully select an appropriate rank for policies in MDPs with uncertain dynamics. Although Theorem 1 provides useful insights, in practice it is difficult to explicitly solve this tradeoff and obtain the optimal rank. To address this, we adopt a greedy strategy: starting from a high-rank

---

**Algorithm 1:** *Adaptive Rank Representation (AdaRL)*

---

**Input:** Initialize parameters: for state-action value $\omega^0$ and policy $\eta^0$. Truncation threshold $\beta \in (0, 1)$, and truncate interval $d_t$.
  **for** $k = 0, 1, \ldots, K - 1$ **do**
    **Data Sampling:** Sample trajectories $\tau_1, \ldots, \tau_N$ from the current policy $\pi_\eta^k$ ,and add them to the replay buffer: $D \leftarrow D \cup \{\tau_1, \ldots, \tau_N\}$
    **Policy Evaluation:** Compute $Q_\omega^k(\cdot, \cdot)$ with sampled data $D$.
    **Policy Improvement:** $\pi_\eta^{k+1}(\cdot|s) \propto \exp(Q_\omega^k(s, \cdot)), \forall s \in \mathcal{S}$.
    **Rank Adaptation Step:** if $k \% d_t = 0$, Search the suitable rank by Eq. 15 and project $\eta_k$ into a lower rank manifold $\mathcal{M}_{\hat{r}}$.
  **end for**

---

model, we gradually reduce the rank until reaching a stable value that yields consistent performance under model uncertainty. This procedure operationalizes the bias–variance tradeoff characterized in Theorem 1 and forms the core of the **Rank Adaptation Step** in our algorithm.

Specifically, the upper-level problem Eq. 13 requires us to identify suitable representations for both the policy and value models while keeping their ranks as low as possible. If no lower-rank model with sufficient approximation quality can be found, we simply retain the previous rank, i.e., $r_{\text{new}} = r_{\text{old}}$. To do the greedy search, we consider using the following criterion to decide the new rank $\hat{r}$. Note there are many ways to decide the target rank; in the ablation study, we show that using this criterion achieves a smooth truncation and makes the rank converge to the intrinsic rank of the environment.

$$\hat{r} = \max\{\ell \in \{1, 2, \ldots, d\} : \frac{\sum_{i=1}^{\ell} \sigma_i}{\sum_{i=1}^{d} \sigma_i} \leq \beta\} \tag{15}$$

To implement this efficiently, we adopt a low-rank factorization approach (Xu et al., 2019; Zhang et al., 2015) that operates directly on the weight matrices of neural networks. Since the rank of a neural network layer is inherently constrained by the number of hidden units, we follow the idea of inserting an intermediate linear layer between consecutive layers, thereby controlling the rank through the size of this hidden layer (see Figure 5 and Appendix A.5.1 for details). After obtaining the optimized policy $\pi_\eta^k$ from several policy optimization steps, we refine this low-rank representation by performing SVD.

**Policy Optimization Step** One can adopt the standard approaches, such as the well-known soft actor critic (SAC) (Haarnoja et al., 2018) algorithm to obtain an approximate optimal policy that solves Eq. 14. Notice that after reconstructing the neural network, the rank of parameter $\eta$ is no larger than $\hat{r}$ due to the existence of the intermediate layer. In this way, the rank constraint is automatically enforced during optimization without requiring explicit SVD at every update.

We summarize the proposed algorithm in Algorithm 1, corresponding to the rank adaptation step and policy improvement step. We conclude this section with a brief remark on the advantages of AdaRL.

**Low Computational Complexity** Unlike previous methods (Gehring et al., 2015) that require repeated SVD with complexity $\mathcal{O}(d^3)$ per update, AdaRL uses a single rank adaptation step to estimate a feasible rank. It operates on two timescales: the inner loop optimizes policies under a low-rank constraint, while the outer loop infrequently adjusts the rank by projecting parameters onto a lower-rank manifold. This design avoids costly worst-case value estimation and yields an efficient training procedure.

**Convergence** Control dynamical systems governed by Newtonian mechanics naturally exhibit a low-rank structure (Tiwari et al., 2025). Although deriving theoretical convergence guarantees is nontrivial, our experiments empirically show that the solution of the upper-level problem converges to a stable rank, thereby balancing model robustness with representational capacity.

## 5 EXPERIMENT

In this section, we present numerical evaluations of the proposed method AdaRL (Alg. 1) and compare it against several robust RL baselines, including RNAC, Parseval regularization, fixed-rank SAC, and

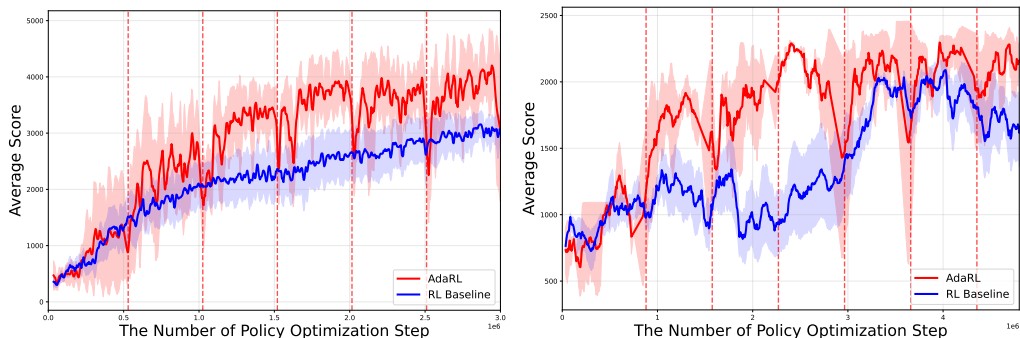

Figure 3: Training performance on MuJoCo tasks. The proposed AdaRL consistently outperforms standard SAC baselines under model uncertainty. The red dashed vertical lines indicate the boundaries between different iteration intervals.

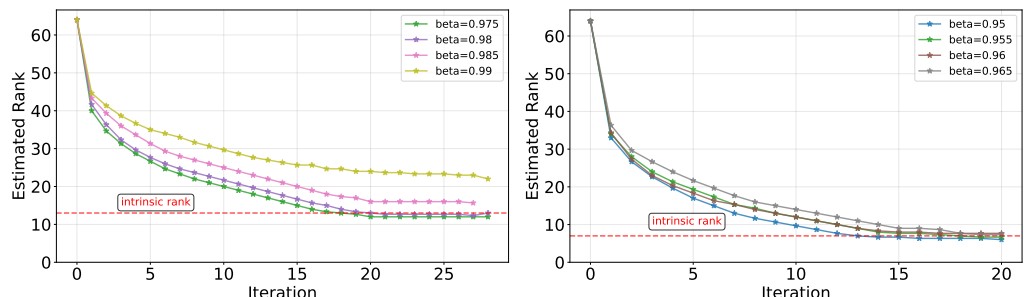

Figure 4: We plot the estimated rank from AdaRL throughout training. The intrinsic rank refers to the value identified by Tiwari et al. (2025). **Left:** Walker2d. **Right:** Hopper.

the algorithm from Tiwari et al. (2025). Our experiments highlight the advantages of AdaRL in two key aspects: (1) it achieves a favorable trade-off between the bias and variance induced by model uncertainty, thereby enabling more robust policy learning; and (2) it identifies a suitable low-rank manifold, within which constraining the policy model yields a representation that remains robust under model uncertainty. More details are given in the Appendix A.5.1.

We focus on robotic control tasks with continuous action spaces, using four widely adopted OpenAI Gym environments and their variants: `Hopper-v3`, `Walker2d-v3`, `Ant-v3`, and `Humanoid-v3`. Following the setup in Luo et al. (2024), we introduce model uncertainty by modifying the source dynamics for each task. In Hopper and Walker2d, this involves structural changes such as adjusting torso and foot sizes, while in Ant and Humanoid we alter physical parameters including gravity or add external forces such as wind with a specified velocity. During training, the environment dynamics vary across episodes to simulate epistemic uncertainty.

The baselines considered in this scenario are: (1) SAC (Haarnoja et al., 2018) with a fixed-rank parameterization; (2) RNAC-DS (Zhou et al., 2023), which employs double sampling within newly defined uncertainty sets and uses function approximation to solve the robust Bellman equation; We also evaluate the RNAC-IPM variant to ensure a more comprehensive and fair comparison. (3) Parseval regularization (Chung et al., 2024), which enforces orthogonality in weight matrices to preserve optimization properties and improve training stability in continual reinforcement learning; and (4) the method of Tiwari et al. (2025), which incorporates a fully connected sparsification MLP layer for reinforcement learning.

In Figure 3 and Table 1, we report numerical results comparing the proposed AdaRL algorithm with several baselines. As shown in Figure 3, both AdaRL and standard SAC achieve similar performance in the first iteration; however, once the model rank is adjusted, AdaRL consistently outperforms the standard methods by mitigating the impact of model uncertainty. It is worth noting that immediately after each rank adaptation step, the optimizer's momentum is reset and the model must adjust to the new parameterization, leading to a temporary performance drop before recovery. Further, in Table 1, the results show that AdaRL consistently outperforms the baselines by a significant margin in most

scenarios. As discussed in Section 2.2, robust RL algorithms typically perform policy improvement based on worst-case value functions, which enhances robustness but often yields overly conservative policies and incurs high approximation errors in continuous control environments (Mannor et al., 2012; 2016; Xu & Mannor, 2012). For regularization-based approaches, Parseval regularization can partially mitigate value-function overfitting, but it remains less effective than the low-rank constraint imposed in AdaRL. To fairly assess policy generalization, all evaluations are conducted under the fixed nominal dynamics $\mathcal{P}^\circ$, enabling us to examine whether the learned policies remain effective and robust in the presence of model uncertainty. In Appendix A.5.4, we further demonstrate the robustness of the trained policy in different perturbed dynamics.

| Task | AdaRL (proposed) | RNAC-DS | RNAC-IPM | Parseval | Alg. in Tiwari et al. (2025) |
|------|------------------|---------|----------|----------|------------------------------|
| Hopper | $\mathbf{2109.8 \pm 322.90}$ | $1542.36 \pm 62.73$ | $1666.33 \pm 495.82$ | $1410.64 \pm 456.52$ | $1850.09 \pm 234.98$ |
| Walker | $\mathbf{3991.90 \pm 567.00}$ | $1906.68 \pm 620.90$ | $2725.42 \pm 570.29$ | $2368.26 \pm 1346.67$ | $3280.55 \pm 179.42$ |
| Ant | $\mathbf{3067.13 \pm 111.55}$ | $1021.97 \pm 230.71$ | $1827.77 \pm 237.64$ | $2063.18 \pm 381.27$ | $2719.95 \pm 225.11$ |
| Humanoid | $\mathbf{5428.72 \pm 50.10}$ | $2351.42 \pm 443.12$ | $3321.49 \pm 342.31$ | $458.35 \pm 76.41$ | $5255.03 \pm 757.45$ |

Table 1: **MuJoCo Results.** The performance of the benchmark algorithms. Bolded numbers indicate the best results among AdaRL, RNAC-DS, RNAC-IPM, Parseval regularization, and the algorithm in Tiwari et al. (2025) for each task.

In Figure 4, we report an additional experiment showing that the rank estimated by the AdaRL algorithm in Eq. 13 gradually converges to the intrinsic rank identified by Tiwari et al. (2025), given an appropriate choice of $\beta$ in Alg. 1 (set to $0.98$ in our experiments). This result demonstrates that AdaRL can effectively search for a suitable rank for environment with model uncertainty.

## 6 CONCLUSION

In this paper, we propose a novel framework for reinforcement learning under epistemic uncertainty by integrating the low-rank structure into policy representation. We begin by establishing a theoretical bias-variance trade-off that arises when applying low-rank approximations with uncertain dynamics. Motivated by this insight, we formulate a bi-level optimization problem and develop the Adaptive Low-Rank Representation algorithm, which dynamically adjusts the policy's representational rank to balance generalization and robustness. Our extensive experiments on MuJoCo benchmarks demonstrate that AdaRL consistently outperforms both fixed-rank RL methods and state-of-the-art robust RL algorithms.

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

## A  APPENDIX

### A.1  LLM USAGE

In this work, LLM were used solely for polishing the writing. No part of the technical content, experimental design, or analysis relied on LLMs. The responsibility for the correctness and originality of the ideas, methods, and results remains entirely with the authors.

### A.2  PROOF OF THEOREM 1

For the ease of notation, we denote the gap between the sampled system dynamics with uncertainty $A_{\mathcal{P}}$ and the reference system $A_{\mathcal{P}^\circ}$ as $\epsilon_A := A_{\mathcal{P}^\circ} - A_{\mathcal{P}}$. Recall that $A_{\mathcal{P},r}$ denotes the low-rank manifold projection of $A_{\mathcal{P}}$ using truncated SVD. Let $A_{\mathcal{P}^\circ}^\dagger$ and $A_{\mathcal{P},r}^\dagger$ respectively denote the pseudo-inverses. To ease notation we write $b_{\mathcal{P}^\circ} := b_{\mathcal{P}^\circ,\omega^\circ}$ and $b_{\mathcal{P}} := b_{\mathcal{P},\omega_{\mathcal{P}}}$. With $\theta^\circ = A_{\mathcal{P}^\circ}^\dagger b_{\mathcal{P}^\circ}$ and

$\theta_r = A_{\mathcal{P},r}^\dagger b_{\mathcal{P}}$, the difference can then be written as:

$$\theta_r - \theta^\circ = A_{\mathcal{P},r}^\dagger b_{\mathcal{P}} - \theta^\circ$$

$$= A_{\mathcal{P},r}^\dagger (b_{\mathcal{P}} - b_{\mathcal{P}^\circ}) + A_{\mathcal{P},r}^\dagger A_{\mathcal{P}^\circ}\theta^\circ - \theta^\circ$$

$$= A_{\mathcal{P},r}^\dagger (b_{\mathcal{P}} - b_{\mathcal{P}^\circ}) + A_{\mathcal{P},r}^\dagger A_{\mathcal{P}}\theta^\circ + A_{\mathcal{P},r}^\dagger \epsilon_A \theta^\circ - \theta^\circ$$

$$= A_{\mathcal{P},r}^\dagger (b_{\mathcal{P}} - b_{\mathcal{P}^\circ}) + \sum_{i=1}^{r} v_i v_i^T \theta^\circ - \sum_{i=1}^{d} v_i v_i^T \theta^\circ + A_{\mathcal{P},r}^\dagger \epsilon_A \theta^\circ$$

$$= A_{\mathcal{P},r}^\dagger (b_{\mathcal{P}} - b_{\mathcal{P}^\circ}) - \sum_{i=r+1}^{d} v_i v_i^T \theta^\circ + A_{\mathcal{P},r}^\dagger \epsilon_A \theta^\circ$$

where the fourth equation above follows from the fact that:

$$A_{\mathcal{P},r}^\dagger A_{\mathcal{P}} = V\Sigma_{\mathcal{P},r}^{-1} U^\top U \Sigma_{\mathcal{P}} V^\top = \sum_{i=1}^{r} v_i v_i^T \quad \text{and} \quad \sum_{i=1}^{d} v_i v_i^T = I.$$

Since the feature functions $\psi, \phi$ are Lipschitz with constant $L > 0$, and that the uncertainty in environment dynamics are bounded from the underlying reference system with Wasserstein distance $W(\hat{\mathcal{P}}_{s,a}^\circ, P_{s,a}) \leq \epsilon$, all the components of matrix $A_{\mathcal{P}}$, say for example $\mathbb{E}_{\mathcal{P}}[\psi(s')\psi(s')^\top]$, can be upper bounded as follows,

$$\sup_{\mathcal{P}\in\mathcal{B}_W(\hat{\mathcal{P}}^\circ,\epsilon)} \left\| \mathbb{E}_{\mathcal{P}}[\psi(s')\psi(s')^\top] - \mathbb{E}_{\hat{\mathcal{P}}^\circ}[\psi(s')\psi(s')^\top] \right\| = \mathcal{O}(L\epsilon)$$

where $\mathcal{B}_W(\hat{\mathcal{P}}^\circ, \epsilon)$ is the Wassertein ball with radius $\epsilon > 0$. Assume $\mathcal{P}^\circ \in \mathcal{B}_W(\hat{\mathcal{P}}^\circ, \epsilon)$, hence by triangle inequality:

$$\left\| \mathbb{E}_{\mathcal{P}}[\psi(s')\psi(s')^\top] - \mathbb{E}_{\mathcal{P}^\circ}[\psi(s')\psi(s')^\top] \right\| \leq 2\mathcal{O}(L\epsilon)$$

It follows that:

$$\|\theta_r - \theta^\circ\|_2 \leq \|A_{\mathcal{P},r}^\dagger (b_{\mathcal{P}} - b_{\mathcal{P}^\circ})\|_2 + \left\| \sum_{i=r+1}^{d} v_i v_i^T \theta^\circ \right\|_2 + 2\mathcal{O}(L\epsilon)$$

$$\leq \frac{1}{\sigma_{\mathcal{P},r}} \|(b_{\mathcal{P}} - b_{\mathcal{P}^\circ})\|_2 + \left\| \sum_{i=r+1}^{d} v_i v_i^T \theta^\circ \right\|_2 + 2\mathcal{O}(L\epsilon)$$

For the second term, we use $\theta^\circ = V^\circ \Sigma_{\mathcal{P}^\circ}^{-1} U^{\circ -1} b_{\mathcal{P}^\circ,\omega}$ to get:

$$\left\| \sum_{i=r+1}^{d} v_i v_i^\top \theta^\circ \right\|_2 = \left\| \sum_{i=r+1}^{d} v_i \sum_{j=1}^{r^\circ} v_i^\top v_j^\circ \sigma_{\mathcal{P}^\circ,j}^{-1} u_j^{\circ\top} b_{\mathcal{P}^\circ} \right\|_2$$

$$\leq \left\| \sum_{i=r+1}^{d} \sum_{j=1}^{r^\circ} v_i^\top v_j^\circ \sigma_{\mathcal{P}^\circ,j}^{-1} u_j^{\circ\top} b_{\mathcal{P}^\circ} \right\|_2$$

$$= \sum_{i=r+1}^{d} \left\| v_i^\top v_i^\circ \sigma_{\mathcal{P}^\circ,i}^{-1} u_i^{\circ\top} b_{\mathcal{P}^\circ} + \sum_{j\neq i}^{r^\circ} v_i^\top v_j^\circ \sigma_{\mathcal{P}^\circ,j}^{-1} u_j^{\circ\top} b_{\mathcal{P}^\circ} \right\|_2$$

$$\leq \sum_{i=r+1}^{d} \|v_i^\top\|_2 \|v_i^\circ\|_2 \sigma_{\mathcal{P}^\circ,i}^{p-1} + \sum_{i=r+1}^{d} \sum_{j\neq i}^{r^\circ} \|v_i^\top v_j^\circ \sigma_{\mathcal{P}^\circ,j}^{-1} u_j^{\circ\top} b_{\mathcal{P}^\circ}\|_2$$

$$\leq (d-r)\sigma_{\mathcal{P}^\circ,r}^{p-1} + \sum_{i=r+1}^{d} \sum_{j\neq i}^{r^\circ} \|v_i^\top v_j^\circ\|_2 \sigma_{\mathcal{P}^\circ,j}^{p-1}$$

$$\leq (d-r)\sigma_{\mathcal{P}^\circ,r}^{p-1} + (d-r)r^\circ \sigma_{\mathcal{P}^\circ,1}^{p-1}$$

It follows that:

$$\|\theta_r - \theta^\circ\|_2 \leq \frac{1}{\sigma_{\mathcal{P},r}}\|b_{\mathcal{P}} - b_{\mathcal{P}^\circ}\|_2 + (d-r)\sigma_{\mathcal{P}^\circ,r}^{p-1} + (d-r)r^\circ\sigma_{\mathcal{P}^\circ,1}^{p-1} + 2\mathcal{O}(L\epsilon) \qquad (16)$$

### A.3 BIAS-VARIANCE IN DEEP RL: THE NEURAL TANGENT KERNEL (NTK) REGIME

In what follows we consider bias-variance decomposition when the neural network is arbitrarily wide (i.e., large number of neurons per layer). In this regime, the network's predictions evolve in a way that can be analytically characterized by the Neural Tangent Kernel (NTK) (Jacot et al., 2018).

#### A.3.1 THE NEURAL TANGENT KERNEL NTK REGIME

Assume $Q_\theta(s,a)$ is represented with a sufficiently wide neural network (NTK regime). In this case:

$$Q_\theta(s,a) \approx Q_{\bar{\theta}}(s,a) + \nabla Q_{\bar{\theta}}(s,a)^\top(\theta - \bar{\theta}),$$

for initialization $\bar{\theta}$. Let

$$\delta(s,a,s';\theta) := Q_\theta(s,a) - R(s,a) - \gamma V_\theta(s')$$

denote the Bellman error where $V_\theta(s) = \log\left(\sum_{a'}\exp Q_\theta(s,a')\right)$. Using the linear approximation we obtain:

$$\delta(s,a,s';\theta) \approx \delta(s,a,s';\bar{\theta}) + \Psi(s,a,s')^\top(\theta - \bar{\theta}),$$

where

$$\Psi(s,a,s') := \nabla Q_{\bar{\theta}}(s,a) - \gamma\sum_{a'}\pi_{\bar{\theta}}(a'|s')\nabla Q_{\bar{\theta}}(s',a'),$$

and $\pi_{\bar{\theta}}$ is the softmax policy induced by $Q_{\bar{\theta}}$. Hence, we minimize the approximated Bellman error:

$$\min_\theta \mathbb{E}_{(s,a,s')\sim\mathcal{P}^\circ}\left[\left(\Psi(s,a,s')^\top(\theta - \bar{\theta}) + \delta(s,a,s';\bar{\theta})\right)^2\right].$$

where $\mathcal{P}^\circ$ be the true joint distribution of $(s,a,s')$. The first order condition can be written as

$$A_{\mathcal{P}^\circ}(\theta - \bar{\theta}) = b_{\mathcal{P}^\circ}$$

where

$$A_{\mathcal{P}^\circ} := \mathbb{E}_{(s,a,s')\sim\mathcal{P}^\circ}\left[\Psi(s,a,s')\Psi(s,a,s')^\top\right] \quad b_{\mathcal{P}^\circ} := -\mathbb{E}_{(s,a,s')\sim\mathcal{P}^\circ}\left[\Psi(s,a,s')\delta(s,a,s';\bar{\theta})\right]$$

The minimum-norm solution is

$$\theta^\circ - \bar{\theta} = A_{\mathcal{P}^\circ}^\dagger b_{\mathcal{P}^\circ} \qquad (17)$$

where $A_{\mathcal{P}^\circ}^\dagger$ is the pseudo-inverse.

#### A.3.2 WASSERSTEIN AMBIGUITY AND PERTURBED BELLMAN–NTK OPERATOR

Let $\hat{\mathcal{P}}$ be an estimated transition model and consider a Wasserstein ball

$$\mathcal{B}(\hat{\mathcal{P}},\varepsilon) := \{\mathcal{P} : W_1(\mathcal{P},\hat{\mathcal{P}}) \leq \varepsilon\},$$

with $\mathcal{P}^\circ \in \mathcal{B}(\hat{\mathcal{P}},\varepsilon)$. Let $\mathcal{P}$ be drawn (uniformly randomly) from $\mathcal{B}(\hat{\mathcal{P}},\varepsilon)$. Bellman error minimization is written as:

$$\min_\theta \mathbb{E}_{(s,a,s')\sim\mathcal{P}}\left[\left(\Psi(s,a,s')^\top(\theta - \bar{\theta}) + \delta(s,a,s';\bar{\theta})\right)^2\right].$$

The first order condition can be written as

$$A_{\mathcal{P}}(\theta - \bar{\theta}) = b_{\mathcal{P}} \qquad (18)$$

where the perturbed Bellman–NTK operator and right-hand side are defined as:

$$A_{\mathcal{P}} := \mathbb{E}_{(s,a,s')\sim\mathcal{P}}\left[\Psi(s,a,s')\Psi(s,a,s')^\top\right], \qquad b_{\mathcal{P}} := -\mathbb{E}_{(s,a,s')\sim\mathcal{P}}\left[\Psi(s,a,s')\bar{\delta}(s,a,s')\right],$$

Consider a truncated singular value decomposition:

$$A_{\mathcal{P},r} = U\Sigma_{\mathcal{P},r}V^\top, \qquad \Sigma_{\mathcal{P},r} = \text{diag}(\sigma_{\mathcal{P},1},\ldots,\sigma_{\mathcal{P},r},0,\ldots,0),$$

with singular values $\sigma_{\mathcal{P},1} \geq \cdots \geq \sigma_{\mathcal{P},r} > 0$. The truncated solution to equation 18 is:

$$\theta_r - \bar{\theta} = A_{\mathcal{P},r}^\dagger b_{\mathcal{P}} \qquad (19)$$

### A.3.3    BIAS-VARIANCE DECOMPOSITION WITH RANK-$r$ BELLMAN–NTK ESTIMATOR

Let $\epsilon_A := A_{\mathcal{P}^\circ} - A_{\mathcal{P}}$ and $d := \text{Rank}(A_{\mathcal{P}})$. It follows that:

$$
\begin{aligned}
\theta_r - \theta^\circ &= A_{\mathcal{P},r}^\dagger b_{\mathcal{P}} - \theta^\circ \\
&= A_{\mathcal{P},r}^\dagger (b_{\mathcal{P}} - b_{\mathcal{P}^\circ}) + A_{\mathcal{P},r}^\dagger A_{\mathcal{P}^\circ} \theta^\circ - \theta^\circ \\
&= A_{\mathcal{P},r}^\dagger (b_{\mathcal{P}} - b_{\mathcal{P}^\circ}) + A_{\mathcal{P},r}^\dagger A_{\mathcal{P}} \theta^\circ + A_{\mathcal{P},r}^\dagger \epsilon_A \theta^\circ - \theta^\circ \\
&= A_{\mathcal{P},r}^\dagger (b_{\mathcal{P}} - b_{\mathcal{P}^\circ}) + \sum_{i=1}^{r} v_i v_i^T \theta^\circ - \sum_{i=1}^{d} v_i v_i^T \theta^\circ + A_{\mathcal{P},r}^\dagger \epsilon_A \theta^\circ \\
&= A_{\mathcal{P},r}^\dagger (b_{\mathcal{P}} - b_{\mathcal{P}^\circ}) - \sum_{i=r+1}^{d} v_i v_i^T \theta^\circ + A_{\mathcal{P},r}^\dagger \epsilon_A \theta^\circ \qquad (20)
\end{aligned}
$$

where the fourth equation above follows from the fact that:

$$
A_{\mathcal{P},r}^\dagger A_{\mathcal{P}} = V \Sigma_{\mathcal{P},r}^{-1} U^\top U \Sigma_{\mathcal{P}} V^\top = \sum_{i=1}^{r} v_i v_i^T \quad \text{and} \quad \sum_{i=1}^{d} v_i v_i^T = I.
$$

Since $A_{\mathcal{P}}$ is Lipschitz (with constant $L > 0$) in $\mathcal{P} \in \mathcal{B}_W(\hat{\mathcal{P}}, \epsilon)$ we have

$$
\|\epsilon_A\|_2 = \|A_{\mathcal{P}} - A_{\mathcal{P}^\circ}\|_2 \leq \|A_{\mathcal{P}} - A_{\hat{\mathcal{P}}}\|_2 + \|A_{\hat{\mathcal{P}}} - A_{\mathcal{P}^\circ}\|_2 \leq 2L\varepsilon
$$

for some $L > 0$. It follows from equation 20 that:

$$
\|\theta_r - \theta^\circ\|_2 \leq \|A_{\mathcal{P},r}^\dagger (b_{\mathcal{P}} - b_{\mathcal{P}^\circ})\|_2 + \Big\| \sum_{i=r+1}^{d} v_i v_i^T \theta^\circ \Big\|_2 + \|A_{\mathcal{P},r}^\dagger \epsilon_A \theta^\circ\|_2.
$$

Using

$$
\|A_{\mathcal{P},r}^\dagger\|_2 \leq \frac{1}{\sigma_{\mathcal{P},r}}, \qquad \|\epsilon_A\|_2 \leq 2L\varepsilon,
$$

we get the bound

$$
\|\theta_r - \theta^\circ\|_2 \leq \frac{1}{\sigma_{\mathcal{P},r}} \|b_{\mathcal{P}} - b_{\mathcal{P}^\circ}\|_2 + \Big\| \sum_{i=r+1}^{d} v_i v_i^T \theta^\circ \Big\|_2 + \frac{2L\varepsilon}{\sigma_{\mathcal{P},r}} \|\theta^\circ\|_2
$$

Since $b_{\mathcal{P}}$ is Lipschitz in $\mathcal{P}$ with constant $L_b$, then

$$
\|b_{\mathcal{P}} - b_{\mathcal{P}^\circ}\|_2 \leq 2L_b \varepsilon,
$$

because $\mathcal{P}, \mathcal{P}^\circ \in \mathcal{B}(\hat{\mathcal{P}}, \varepsilon)$. The upper bound can be expressed as:

$$
\|\theta_r - \theta^\circ\|_2 \leq \underbrace{\frac{2\varepsilon(L_b + L\|\theta^\circ\|_2)}{\sigma_{\mathcal{P},r}}}_{\text{variance}} + \underbrace{\Big\| \sum_{i=r+1}^{d} v_i v_i^\top \theta^\circ \Big\|_2}_{\text{bias}} \qquad (21)
$$

The rank-$r$ truncation suppresses high-variance, low-signal directions of the Bellman–NTK operator, improving stability under epistemic uncertainty in dynamics, while introducing bias due to discarded spectral components. This reveals an explicit bias–variance tradeoff governed by the singular value spectrum of $A_{\mathcal{P}}$

### A.4    HEURISTIC ARGUMENT FOR CONVERGENCE OF ADARL

This section provides a geometric interpretation of the proposed adaptive-rank bi-level method, and gives a heuristic argument for the stability and convergence behavior observed empirically. While a full global convergence theorem is beyond the scope of the present work—given the combination of deep function approximation, bi-level structure, and rank adaptation—the algorithm exhibits a well-organized structure that allows for a clear explanation of why the rank stabilizes and why the method behaves like a conventional actor–critic algorithm thereafter (Tian et al., 2023; Dong et al., 2022; Fu et al., 2020).

**Low-Rank Parameter Space as a Determinantal Variety**    Let $\theta \in \mathbb{R}^{d_1 \times d_2}$ denote the matrix of policy parameters. For a fixed maximal rank $\bar{r}$, define the determinantal variety

$$\mathcal{M}_{\leq \bar{r}} := \{\theta \in \mathbb{R}^{d_1 \times d_2} : \text{rank}(\theta) \leq \bar{r}\},$$

which is a closed, semi-algebraic subset of $\mathbb{R}^{d_1 \times d_2}$ stratified by smooth manifolds

$$\mathcal{M}_r := \{\theta : \text{rank}(\theta) = r\}, \qquad r = 0, 1, \ldots, \bar{r}.$$

The singular locus of $\mathcal{M}_{\leq \bar{r}}$ consists of lower-rank strata, but this does not affect the argument below. Algorithm 1 ensures that *all iterates* $\theta^k$ lie in $\mathcal{M}_{\leq \bar{r}}$, so parameter updates occur inside a single algebraic variety.

**Monotone and Finite Rank Adaptation**    The Rank Adaptation Step in Algorithm 1 employs the truncation rule

$$\hat{r} = \max \left\{ \ell \in \{1, \ldots, d\} : \frac{\sum_{i=1}^{\ell} \sigma_i}{\sum_{i=1}^{d} \sigma_i} \leq \beta \right\},$$

computed from the singular values $\sigma_1 \geq \sigma_2 \geq \cdots$ of the current parameter matrix. Starting from a high-rank model, the algorithm only *decreases* the rank when the resulting approximation preserves sufficient representation power. Thus the sequence of ranks satisfies

$$r_{k+1} \leq r_k, \qquad r_k \in \{1, \ldots, \bar{r}\}.$$

Since this sequence only takes values in a finite set and is monotonically nonincreasing, it can perform *only finitely many* strict decreases. Therefore there exists a finite iteration $K$ and a rank $\hat{r}$ such that

$$r_k = \hat{r} \quad \text{for all } k \geq K.$$

Beyond iteration $K$, all iterates lie in the fixed-rank manifold $\mathcal{M}_{\hat{r}}$.

**Reduction to Standard Policy Optimization**    Once the rank has stabilized, Algorithm 1 reduces to entropy-regularized actor–critic training on the smooth manifold $\mathcal{M}_{\hat{r}}$, implemented via a low-rank bottleneck layer. Thus the bi-level procedure collapses to a standard single-level policy-gradient method with a fixed structured parameterization and thus converges.

This behavior matches the empirical observations in Section 5: the rank changes only a small number of times, after which the robust return and policy iterates stabilize.

**Summary**    Although a formal global convergence theorem is not provided, the algorithm exhibits the following structured behavior:

- All iterates remain in the determinantal variety $\mathcal{M}_{\leq \bar{r}}$.
- The rank adaptation mechanism is monotone and thus stabilizes after finitely many updates.
- After stabilization at rank $\hat{r}$, the method reduces to standard entropy-regularized policy optimization on the smooth manifold $\mathcal{M}_{\hat{r}}$.

This geometric viewpoint explains why AdaRL converges to a stable rank and why subsequent training behaves like conventional actor–critic learning with a fixed low-rank representation.

## A.5    ADDITIONAL RESULT

### A.5.1    BASIC SETTINGS

In all experiments, we evaluate the performance of benchmark algorithms on the `Hopper-v3`, `Walker2d-v3`, `Humanoid-v3`, and `Ant-v3` environments from OpenAI Gym. To ensure a fair comparison, we use the open-source implementation[1] of SAC as the base RL algorithm for all methods, and for RNAC we adopt its original PPO-based trainer without modification. We use Adam as the optimizer in SAC, where both the policy and Q-networks are implemented as two-layer MLPs with hidden sizes $(64, 64)$ and ReLU activation functions. The learning rate for both networks is

---

[1] `https://github.com/openai/spinningup`

fixed at $3 \times 10^{-3}$. For our proposed algorithm, we set the truncation interval $d_t$ to $0.7 \times 10^6$ for Walker2d and $10^6$ for Hopper, Humanoid, and Ant, meaning the model is truncated every $d_t$ policy optimization steps. This choice ensures that rank adaptation occurs much less frequently than policy updates.

To impose a rank constraint on a weight matrix $W$, we first factorize it as $W = W_1 W_2$ and apply singular value decomposition (SVD) to the product $W_1 W_2 = U \Sigma V^\top$. We then reparameterize as

$$W_1 = U_{[:,:\hat{r}]} \sqrt{\Sigma_{[:\hat{r}]}}, \quad W_2 = \sqrt{\Sigma_{[:\hat{r}]}} V_{[:\hat{r},:]},$$

where $\hat{r} \leq r$ is the target rank. This projects $W$ onto a lower-rank manifold, thereby enforcing the constraint. As shown in Figure 5, inserting an intermediate linear layer (yellow, within the red region) provides an explicit implementation of this rank reduction.

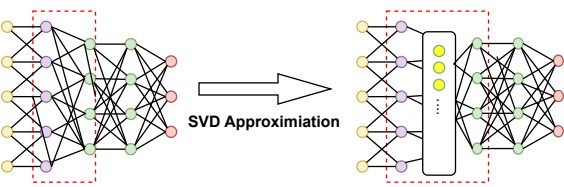

Figure 5: To impose the low-rank constraint, we insert an intermediate linear layer (without activation functions or bias) between the original two layers. This layer acts as a bottleneck that enforces a low-rank factorization of the weight matrix via SVD approximation.

Additionally, to avoid loss of momentum after optimizer resets, we apply a standard cosine decay schedule with warm-up, as in (Lialin et al., 2023; Touvron et al., 2023). Specifically, upon each reset, we set the learning rate to zero, gradually warm it up to the target value over 2000 steps, and then resume following the cosine schedule.

We present the practical implementation of our proposed algorithm in Alg. 1. At each iteration, we warm-start both the policy network and Q-network in SAC using the trained neural networks from the previous iteration, and then run SAC in the corresponding MuJoCo environment to continue training.

For the robust RL baselines, we use their official open-source implementations. The implementation of RNAC is available at `https://github.com/tliu1997/RNAC`. To modify the dynamics kernel, we follow the setting in OMPO Luo et al. (2024), using their codebase at `https://github.com/Roythuly/OMPO`. For Parseval regularization, we use the implementation provided at `https://github.com/wechu/parseval_reg`. In MuJoCo experiments with Parseval regularization, we adopt the same setup, tuning the regularization coefficient from $\{0.001, 0.0001, 0.00001\}$ and selecting the best-performing value. We also follow the original implementation by setting $s = 2$ in the Parseval constraint $\|WW^\top - sI\|_F$. For Tiwari et al. (2025), we follow their default configuration with a sparsification layer and set the hidden layer size to 1024 neurons, consistent with their original setting.

### A.5.2 MODEL UNCERTAINTY SETTING

Following the setup in Luo et al. (2024), we simulate model uncertainty by introducing continuously varying environment parameters during training. This design encourages policies to generalize across dynamic variations rather than overfitting to a fixed set of dynamics. The specific parameter schedules for each environment are as follows:

- **Hopper:** The torso and foot lengths vary with the episode index $i$ as

$$L_{\text{torso}}(i) = 0.4 + 0.2 \cdot \sin(0.2i), \quad L_{\text{foot}}(i) = 0.39 + 0.2 \cdot \sin(0.2i).$$

- **Walker2d:** The torso and foot lengths follow a similar pattern with

$$L_{\text{torso}}(i) = 0.2 + 0.1 \cdot \sin(0.3i), \quad L_{\text{foot}}(i) = 0.1 + 0.05 \cdot \sin(0.3i).$$



Figure 6: Visualization of uncertain dynamics in the Hopper-v3 task, where the torso and foot lengths vary across episodes.

- **Ant:** Gravity $g$ and wind speed $W$ change across episodes according to

$$g(i) = 14.715 + 4.905 \cdot \sin(0.5i), \quad W(i) = 1 + 0.2 \cdot \sin(0.5i).$$

- **Humanoid:** The same variation as Ant is applied, but the wind effect is amplified due to the humanoid's larger mass and drag:

$$g(i) = 14.715 + 4.905 \cdot \sin(0.5i), \quad W(i) = 1 + 0.5 \cdot \sin(0.5i).$$

### A.5.3 RANK CONVERGENCE OF THE ALTERNATIVE ALGORITHM

In this subsection, we conduct an ablation study to examine alternative strategies for selecting the cut-off rank of the SVD beyond Eq. 15. As reviewed by Falini (2022), numerous criteria have been proposed for truncated SVD. Here, we consider a simple hard-thresholding approach based on the ratio between singular values. Specifically, we define the cut-off rank as

$$\hat{r} = \min\left\{ \ell \in \{1, 2, \ldots, d\} \mid \frac{\sigma_\ell}{\sigma_1} \le \beta \right\}. \tag{22}$$

Figure 7 illustrates a fundamental limitation of this criterion. After the initial iteration, the rank selection process stagnates because the rule in Eq. 22 depends only on the largest singular value. As a result, it ignores the broader spectral structure of the parameters and fails to adapt dynamically to spectral variations during training. Therefore, we continue to use Eq. 15 as our primary rule for selecting the cut-off rank.

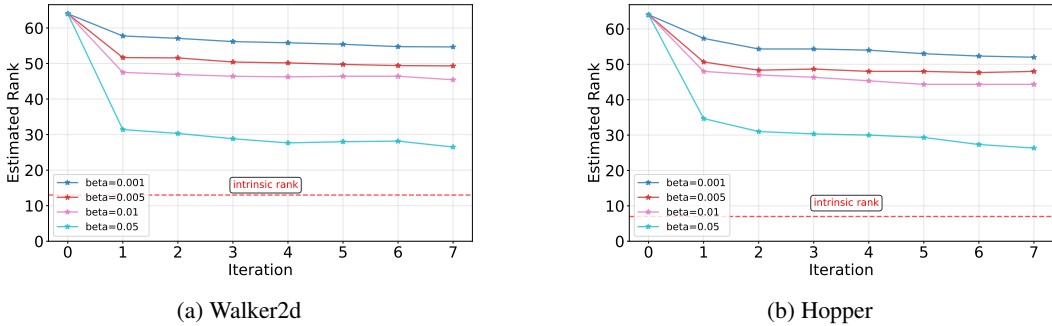

(a) Walker2d          (b) Hopper

Figure 7: Comparison of Rank Selection by hard-thresholding method.

### A.5.4 POLICY PERFORMANCE UNDER VARYING DYNAMICS

In this subsection, we present additional experimental results under perturbations of physical hyper-parameters (e.g., torso length, foot length) in the `Hopper-v3` and `Walker2d-v3` environments. As shown in Figure 8, the proposed AdaRL algorithm exhibits superior robustness and outperforms the strongest baseline (Tiwari et al., 2025) in the majority of cases.

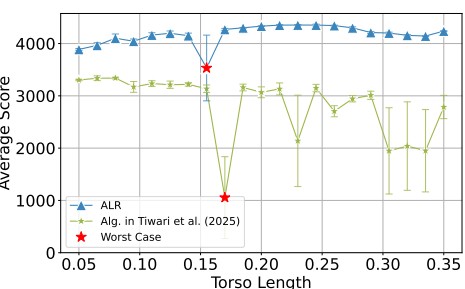 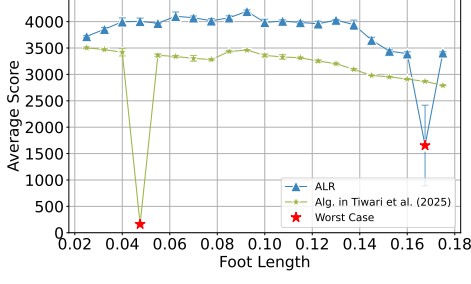

(a) `Walker2d-v3`: Varying torso length with fixed foot length

(b) `Walker2d-v3`: Varying foot length with fixed torso length

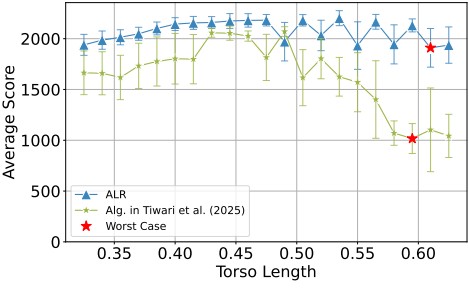 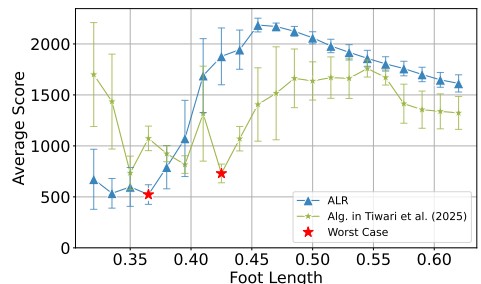

(c) `Hopper-v3`: Varying torso length with fixed foot length

(d) `Hopper-v3`: Varying foot length with fixed torso length

Figure 8: Policy performance under perturbations of physical hyperparameters in `Walker2d-v3` and `Hopper-v3`. Each curve reports the mean performance with shaded regions indicating the standard deviation across seeds. Red pentagram markers (⋆) denote the worst-case performance under each perturbation setting. Subfigures (a) and (c) correspond to varying torso length with fixed foot length, while (b) and (d) show results for varying foot length with fixed torso length. The proposed AdaRL algorithm consistently outperforms the strongest baseline (Tiwari et al., 2025) in most perturbed settings, demonstrating improved robustness.

### A.5.5 ABLATION STUDY OF ADARL

In this section, we report ablations clarifying where the rank constraints are applied. We evaluate three variants of AdaRL: (i) **actor-only**, where only the policy network is re-factorized; (ii) **critic-only**, where only the value network is re-factorized; and (iii) **both**, which corresponds to the full AdaRL method. For each variant, we apply the factorization described in Figure. 5 to the first two layers while keeping all other components identical.

The results in Table 2 show that applying rank adaptation to either the actor alone or both the actor and critic yields substantial robustness improvements across uncertainty levels and environments. In contrast, the critic-only variant consistently provides limited gains and is often the weakest among the three. This pattern suggests that controlling the expressiveness of the policy network is the primary driver of robustness, while jointly adapting both components offers the most reliable and stable performance. Overall, the ablations demonstrate that AdaRL benefits notably from rank adaptation on the actor side, with the actor–critic configuration delivering the strongest results.

### A.5.6 VALIDATING WITH A TOY EXAMPLE IN A WASSERSTEIN BALL

In this section, we provide a numerical CartPole example to verify the theorem. Following the corrected dynamics in Florian (2007), the CartPole system can be written as

$$
\begin{bmatrix} \dot{x} \\ \ddot{x} \\ \dot{\theta} \\ \ddot{\theta} \end{bmatrix} = \begin{bmatrix} 0 & 1 & 0 & 0 \\ 0 & 0 & -\frac{mg}{(m+M)\left(\frac{4}{3}-\frac{m}{m+M}\right)} & 0 \\ 0 & 0 & 0 & 1 \\ 0 & 0 & \frac{g}{l\left(\frac{4}{3}-\frac{m}{m+M}\right)} & 0 \end{bmatrix} \begin{bmatrix} x \\ \dot{x} \\ \theta \\ \dot{\theta} \end{bmatrix} + \begin{bmatrix} 0 \\ \frac{1}{m+M}+\frac{m}{(m+M)^2\left(\frac{4}{3}-\frac{m}{m+M}\right)} \\ 0 \\ -\frac{1}{l(m+M)\left(\frac{4}{3}-\frac{m}{m+M}\right)} \end{bmatrix} u, \quad (23)
$$

Table 2: Performance of the AdaRLvariants across environments and uncertainty levels. Each entry reports the average return over 5 random seeds. Rows highlighted in light blue denote the method achieving the highest performance at Iteration 5.

| Methods | Iteration 1 (0.7e6 Step) | Iteration 3 (2.1e6 Step) | Iteration 5 (3.5e6 Step) |
|---|---|---|---|
| **Hopper-v3 with Low Uncertainty (torso_len $\in [0.31, 0.49]$, foot_len $\in [0.305, 0.485]$)** | | | |
| ALR (Actor Only) | $1308.14 \pm 275.42$ | $2120.85 \pm 115.20$ | $2264.38 \pm 80.59$ |
| ALR (Critic Only) | $1181.78 \pm 197.83$ | $1910.93 \pm 484.25$ | $2076.15 \pm 476.52$ |
| ALR (Actor–Critic Both) | $848.50 \pm 346.82$ | $2205.95 \pm 148.13$ | $2259.59 \pm 40.51$ |
| **Hopper-v3 with High Uncertainty (torso_len $\in [0.25, 0.55]$, foot_len $\in [0.245, 0.545]$)** | | | |
| ALR (Actor Only) | $890.83 \pm 724.68$ | $1614.52 \pm 501.20$ | $1813.32 \pm 223.40$ |
| ALR (Critic Only) | $976.27 \pm 598.22$ | $1778.76 \pm 28.67$ | $1804.34 \pm 255.27$ |
| ALR (Actor–Critic Both) | $608.77 \pm 351.49$ | $1981.51 \pm 214.14$ | $2245.84 \pm 56.32$ |
| **Walker2d-v3 with Low Uncertainty (torso_len $\in [0.1, 0.3]$, foot_len $\in [0.05, 0.15]$)** | | | |
| ALR (Actor Only) | $1780.43 \pm 579.17$ | $3242.00 \pm 247.36$ | $3356.54 \pm 157.38$ |
| ALR (Critic Only) | $1481.17 \pm 692.37$ | $3658.15 \pm 454.64$ | $3699.68 \pm 569.28$ |
| ALR (Actor–Critic Both) | $2171.12 \pm 96.99$ | $4095.72 \pm 83.21$ | $4692.13 \pm 370.46$ |
| **Walker2d-v3 with High Uncertainty (torso_len $\in [0.06, 0.34]$, foot_len $\in [0.03, 0.17]$)** | | | |
| ALR (Actor Only) | $1476.82 \pm 957.33$ | $3644.88 \pm 198.58$ | $3686.93 \pm 160.31$ |
| ALR (Critic Only) | $1253.01 \pm 805.23$ | $3433.09 \pm 189.79$ | $3490.55 \pm 365.96$ |
| ALR (Actor–Critic Both) | $1796.41 \pm 859.72$ | $3146.30 \pm 447.84$ | $3348.37 \pm 270.98$ |

where $x$ denotes the horizontal position of the cart, $\dot{x}$ its velocity, $\theta$ the pole angle (measured from the upright position), and $\dot{\theta}$ the angular velocity. The parameters $m$ and $M$ are the pole and cart masses, respectively, $l$ is the pole length (0.5 in the default setting), $g$ is the gravitational acceleration, and $u \in \{0, 1\}$ represents the control input (horizontal force) applied to the cart.

To inject robustness and model uncertainty into this system, we assume that the pole length $l$ varies across episodes. Let $l_0$ denote the nominal length used to define the reference dynamics. During training, we randomly sample $l$ from the interval

$$l \in [0.95\,l_0,\ 1.05\,l_0].$$

For a fixed pole length $l$, the CartPole dynamics in equation 23 can be written compactly as

$$\dot{s} = A(l)\,s + B(l)\,u, \qquad s = [x, \dot{x}, \theta, \dot{\theta}]^\top,\ u \in \{0, 1\},$$

where $A(l)$ and $B(l)$ are obtained directly from the matrices in equation 23.

Let $l_0$ be the nominal length and denote by

$$s_0^+ = A(l_0)s + B(l_0)u, \qquad s^+(l) = A(l)s + B(l)u$$

the next states under $l_0$ and $l$, respectively. Since the system is deterministic, the transition kernels are Dirac measures

$$\hat{P}_{s,a}^\circ = \delta_{s_0^+}, \qquad P_{s,a}(l) = \delta_{s^+(l)}.$$

For a Wasserstein distance with Euclidean ground cost, we then have

$$W\big(\hat{P}_{s,a}^\circ, P_{s,a}(l)\big) = \big\|s_0^+ - s^+(l)\big\|_2 = \big\|(A(l_0) - A(l))s + (B(l_0) - B(l))u\big\|_2.$$

Assume the state is bounded, $\|s\| \leq \|s\|_{\max}$, and recall that $u \in \{0, 1\}$, hence $|u| \leq 1$. Using operator norms, we obtain

$$W\big(\hat{P}_{s,a}^\circ, P_{s,a}(l)\big) \ \leq \ \|A(l_0) - A(l)\|\,\|s\|_{\max} + \|B(l_0) - B(l)\|.$$

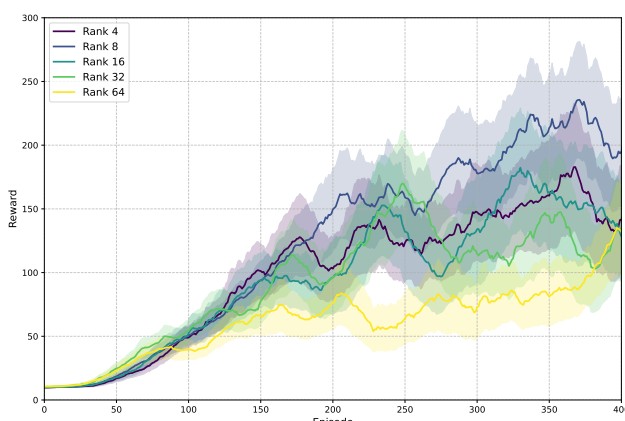

Figure 9: Performance of different-rank policy models under CartPole dynamics uncertainty.

The entries of $A(l)$ and $B(l)$ depend on $l$ only through rational functions such as $\frac{1}{l}$ and $\frac{1}{l(m+M)}$. On the compact interval $l \in [0.95\,l_0, 1.05\,l_0]$ these functions are Lipschitz, so there exist constants $K_A, K_B > 0$ such that

$$\|A(l_0) - A(l)\| \le K_A|l - l_0|, \qquad \|B(l_0) - B(l)\| \le K_B|l - l_0|.$$

Hence, for any $(s,a)$ and any $l \in [0.95\,l_0, 1.05\,l_0]$,

$$W\big(\hat{P}^\circ_{s,a}, P_{s,a}(l)\big) \;\le\; \big(K_A\|s\|_{\max} + K_B\big)\,|l - l_0| \;\le\; 0.05\,l_0\,\big(K_A\|s\|_{\max} + K_B\big).$$

Therefore, by choosing

$$\varepsilon_{s,a} \;:=\; 0.05\,l_0\,\big(K_A\|s\|_{\max} + K_B\big),$$

(or a global $\varepsilon$ using the supremum over $(s,a)$), the perturbed dynamics with $l \in [0.95\,l_0, 1.05\,l_0]$ indeed satisfy

$$P_{s,a}(l) \in \Big\{ P_{s,a} \in \Delta_{\mathcal{S}} \;\Big|\; W\big(\hat{P}^\circ_{s,a}, P_{s,a}\big) \le \varepsilon_{s,a} \Big\},$$

i.e., they lie inside a Wasserstein ball around the nominal transition kernel.

**Numerical Results** Following the setup in Section 4.2, we conduct a numerical experiment to examine whether the model rank affects the performance of this linear control system. We perform a sanity check using models of different ranks (4, 8, 16, 32, 64). As shown in Figure 9, although the nominal CartPole dynamics suggest an optimal rank of 4 (Equation 23), introducing model uncertainty requires greater capacity, and the model with rank = 8 achieves the best performance—closely aligning with our theoretical prediction.

