# OpenReview forum: "ADARL: Adaptive Low-Rank Structures for Robust Policy Learning under Uncertainty"
_ICLR.cc/2026/Conference — ICLR 2026 Conference Desk Rejected Submission_

### Official Review · Reviewer_ucxQ · 2025-10-30

**Soundness:** 2
**Presentation:** 2
**Contribution:** 2
**Rating:** 4
**Confidence:** 3

**Summary:**

The paper introduces AdaRL (Adaptive Rank Representation for Reinforcement Learning), a novel framework for robust RL under epistemic uncertainty. Traditional robust RL methods rely on nested min–max optimization, which is computationally expensive and tends to produce overly conservative policies. AdaRL instead leverages adaptive low-rank policy representations to align model complexity with the intrinsic dimensionality of the task.

**Strengths:**

* The bias–variance trade-off (Theorem 1) under epistemic uncertainty is well-motivated and technically non-trivial.
* The low-rank structure itself as a helpful prior can be useful to handle the uncertainty in the system. Interstingly, the results of the low-rank policy correspond to the rank of the task.

**Weaknesses:**

* The main concern lies in the paper’s positioning. The authors claim that AdaRL is a principled alternative for robust RL under model uncertainty; however, it appears to be more accurately described as a method for solving (soft) robust RL problems rather than redefining the framework itself.
* The proposed method seems somewhat disconnected from the experimental design. While the algorithm assumes uniform sampling from a Wasserstein uncertainty set over dynamics, the experiments rely on predefined modifications to system parameters. This discrepancy weakens the rigour of the robustness claims.

* There is still room for improvement in presentation and clarity:

  - Figure 1 is blurry and mostly reiterates text descriptions; the framework could be visualized more intuitively.
  - Line 113: it should be **P** instead of **\mathcal{P}** in the MDP tuple.
  - $\theta$ is the parameter of the Q-function of the policy function?
  - Theorem 1, can you further explain what is $b_{\mathcal{P},\omega^o}$  since in Equation 8 $\omega^o$ should be calculated based on $\mathcal{P}^o$ , and where is $\sigma_{\mathcal{P},r}$ defined?

  - Figure 2 contains text that is too small to read.

  - Figure 3 includes unclear or incomplete titles.

* The experimental evaluation is limited to a small set of tasks and parameter variations. Including comparisons such as training with predefined fixed-rank policies would provide a stronger validation of the adaptive mechanism.

**Questions:**

* Is there a way to mitigate or handle the temporary performance drop observed after rank changes?
* The paper shows that the learned rank converges to the intrinsic rank of each task. Could you briefly explain how this intrinsic rank is defined? Is it a property of the environment, the policy representation, or both?
* In Equation (7), why not explicitly define the expectation over both sss and aaa? The current notation could be clarified for better readability.
* In Equation (11), what is the purpose of the projection operator if Equation (12) already constrains the policy to lie in $\mathcal{M}_r$?

---

> ### Author Response · Authors · 2025-11-21
>
> > The main concern lies in the paper’s positioning. The authors claim that AdaRL is a principled alternative for robust RL under model uncertainty; however, it appears to be more accurately described as a method for solving (soft) robust RL problems rather than redefining the framework itself.
>
> **Response**: We thank the reviewer for this insightful comment and appreciate the opportunity to clarify our positioning.
>
> As discussed in our response to Reviewer niSV, classical robust RL and DRO-based methods fundamentally rely on computing or approximating the worst-case value function over an uncertainty set. While principled, this requires solving a nested min–max problem and explicitly tracking the worst-case MDP—tasks that are known to be intractable in high-dimensional continuous-control settings unless very strong structural assumptions (e.g., convex rectangular ambiguity sets) are imposed.
>
> AdaRL is therefore not intended to approximate the DRO inner maximization, nor is it simply a soft version of classical robust RL. Instead, our method bypasses worst-case evaluation altogether by adopting a different robustness mechanism: adaptive structural regularization. The inner loop estimates the Bellman error under plausible perturbations, while the outer loop adaptively adjusts the policy’s intrinsic rank, which we show is closely tied to robustness and over-parameterization effects.
>
> For this reason, we describe AdaRL as a framework. It reorganizes the robust RL objective into a tractable bi-level procedure that avoids estimating worst-case dynamics, mitigates the computational challenges inherent to DRO-based methods, and provides a practical and principled alternative for achieving robustness in continuous-control domains.
>
>
>
> > The proposed method seems somewhat disconnected from the experimental design. While the algorithm assumes uniform sampling from a Wasserstein uncertainty set over dynamics, the experiments rely on predefined modifications to system parameters. This discrepancy weakens the rigour of the robustness claims.
>
> **Response**: Thank you for the thoughtful comment. In most continuous-control environments, including MuJoCo, the true transition kernel is unknown and cannot be accessed directly. Consequently, constructing a Wasserstein uncertainty set over transition dynamics is not feasible in practice (and the same limitation holds in real-world applications). For this reason, and following common practice in the robust RL literature, we use predefined perturbations to system parameters to approximate uncertainty in the dynamics. This experimental design is widely adopted in prior works [1] and in standard benchmarks [2] (e.g., robust MuJoCo perturbation suites), and it enables controlled, reproducible, and meaningful comparisons across methods.
>
> We have clarified this connection in the revised manuscript and emphasized that our experimental protocol is fully aligned with established evaluation standards for robust RL in continuous-control settings.
>
> Reference: [1] Luo, Yu, et al. "Ompo: A unified framework for rl under policy and dynamics shifts." arXiv preprint arXiv:2405.19080 (2024).
> [2] Gu, Shangding, et al. "Robust gymnasium: A unified modular benchmark for robust reinforcement learning." arXiv preprint arXiv:2502.19652 (2025).
>
> > There is still room for improvement in presentation and clarity:
> > • Figure 1 is blurry and mostly reiterates text descriptions; the framework could be visualized more intuitively.
>
> **Response**: We thank the reviewer for the helpful suggestion. In the revised manuscript, we have reworked Figure 1 to improve both clarity and visual intuitiveness. The figure has been redrawn with a clearer structure and updated annotations, and we have also refined the accompanying description to better highlight the key components of our framework.
>
> > • Line 113: it should be P instead of \(\mathcal{P}\) in the MDP tuple.
>
> **Response**: Thank you for pointing this out. We have corrected the notation.
>
> > • $\theta$ is the parameter of the Q-function of the policy function?
>
> **Response**: Sorry for the confusion. Here, $\theta$ denotes the parameters of the Q-function. We have updated the notation in Section 4 of the revised version to avoid this conflict.

---

> > ### Author Response · Authors · 2025-11-21
> >
> > > • Figure 2 contains text that is too small to read.
> > > • Figure 3 includes unclear or incomplete titles.
> >
> > **Response**: Thank you for pointing this out. We have revised both figures by enlarging the text and clarifying the titles to ensure they are easy to read.
> >
> > > The experimental evaluation is limited to a small set of tasks and parameter variations. Including comparisons such as training with predefined fixed-rank policies would provide a stronger validation of the adaptive mechanism.
> >
> > **Response**: Thank you for the helpful suggestion. We have added experiments comparing AdaRL with predefined fixed-rank policies (SAC). These additional results show that the adaptive mechanism consistently achieves stronger performance across both Hopper-v3 and Walker2d-v3.
> >
> > We would also like to kindly remind the reviewer that AdaRL’s advantage comes precisely from its ability to adaptively select an appropriate rank. As the tables demonstrate, the automatically chosen ranks lead to substantially better outcomes than fixed-rank alternatives.
> >
> > The updated results are provided below. We hope these results help address the reviewer’s concern.
> >
> > | Method          | Hopper-v3 (Mean ± Std) | Walker2d-v3 (Mean ± Std) |
> > |-----------------|----------------------------------|------------------------------------|
> > | SAC (Rank 16)   | 1049.09 ± 802.07                 | 2322.61 ± 949.95                   |
> > | SAC (Rank 24)   | 1774.57 ± 654.12                 | 2116.05 ± 953.08                   |
> > | SAC (Rank 32)   | 2103.70 ± 188.09                 | 3395.98 ± 313.25                   |
> > | SAC (Rank 40)   | 1744.15 ± 550.67                 | 3316.38 ± 521.52                   |
> > | SAC (Rank 48)   | 1312.06 ± 589.84                 | 2840.69 ± 569.91                   |
> > | AdaRL  | **2109.8 ± 322.90**              | **3991.90 ± 567.00**           |

---

> > > ### Author Response · Authors · 2025-11-21
> > >
> > > Questions:
> > >
> > > > Q1) Is there a way to mitigate or handle the temporary performance drop observed after rank changes?
> > >
> > > **Response**: Thank you for the question. The temporary performance drop after a rank change is primarily caused by the refactorization step: when the rank is modified, the weight matrix is reconstructed, and the optimizer’s internal states (e.g., Adam’s momentum and variance terms) cannot be reused. Consequently, the optimizer needs a brief period to rebuild these statistics, which leads to short-lived instability.
> > >
> > > To mitigate this effect, we already applied a warm-up learning-rate schedule immediately after each rank update (returning to the default learning rate within roughly 100 epochs), which stabilizes training and accelerates recovery. Importantly, this fluctuation is very short and does not affect the subsequent convergence or the final performance. We therefore consider this temporary dip acceptable and an inherent consequence of performing structural changes to the parameterization.
> > > > Q2) The paper shows that the learned rank converges to the intrinsic rank of each task. Could you briefly explain how this intrinsic rank is defined? Is it a property of the environment, the policy representation, or both?
> > >
> > > **Response:** Thank you for the insightful question. The intrinsic rank refers to the minimal number of action directions needed to represent an optimal control strategy for a given task. Following the insight of Theorem 1 from [1], the agent’s attainable state manifold is upper-bounded by $2d_a + 1$ dimensions. This implies that, despite the high nominal action dimensionality, only a small subset of action directions meaningfully influences the system dynamics.
> > >
> > > Consequently, the optimal policy only varies along these effective directions, and its parameterization naturally admits a low-rank structure. Our adaptive method converges to this minimal rank in practice.
> > >
> > > [1] Tiwari, Saket, Omer Gottesman, and George Konidaris. "Geometry of Neural Reinforcement Learning in Continuous State and Action Spaces." The Second Conference on Parsimony and Learning (Recent Spotlight Track). 2025.
> > >
> > > > Q3) In Equation (7), why not explicitly define the expectation over both sss and aaa? The current notation could be clarified for better readability.
> > >
> > > **Response**:Thank you for the suggestion. For clarity, we have updated the notation in Equation (7) accordingly in the revised manuscript.
> > >
> > > > Q4) In Equation (11), what is the purpose of the projection operator if Equation (12) already constrains the policy to lie in \(M_r\)?
> > >
> > > **Response**: Thank you for this valuable observation regarding the typo in our original problem formulation. The lower-level objective should correctly be constrained to the set $\mathbf{\theta \in \mathcal{M}_{\leq \bar{r}}}$. That is, $\theta^*$ parametrizes the optimal policy within the algebraic variety of matrices with rank less than or equal to $\bar{r}$. In the upper level, we formalize an optimal tradeoff between the approximation error (given by the projection operator) and a rank penalty $\mathbf{\lambda r}$. We have corrected this typo in the revised version of the paper.

---

> > > > ### Author Response · Authors · 2025-11-21
> > > >
> > > > > • Theorem 1, can you further explain ..
> > > >
> > > > **Response**: Thanks for your careful comment which highlights a typo in the upper bound expression. We denote by $(\theta^{\circ},\omega^{\circ})$ the solution to the bi-level formulation of entropy regularized RL problem:
> > > >
> > > > \begin{equation}
> > > > \min_{\omega} \quad \mathbb{E}_{\mathcal{P}^{\circ}}\Big[\|\psi(s')^{\top}\omega-\log \sum\limits _ {a'} \exp{\phi(s',a')^{\top} \theta^*(\omega)}\|^2\Big]
> > > > \end{equation}
> > > >
> > > > \begin{equation}
> > > > \text{s.t.} \quad \theta^*(\omega)=\arg \min_{\theta}\mathbb{E}_{ \mathcal{P}^{\circ}}\Big[\|R(s,a)+\gamma \psi(s')^\top \omega-\phi(s, a)^\top \theta\|^2\Big]
> > > > \end{equation}
> > > >
> > > > As described in the paper, the first order condition implies that
> > > > $$
> > > > \theta^\circ
> > > > = A_{\mathcal{P}^\circ}^\dagger b_{\mathcal{P}^\circ,\omega^{\circ}}
> > > > $$ Similarly, we consider the perturbed version of the problem:
> > > >
> > > > \begin{equation}
> > > > \min_{\omega} \quad \mathbb{E}_{\mathcal{P}}\Big[\|\psi(s')^{\top}\omega-\log \sum\limits _ {a'} \exp{\phi(s',a')^{\top} \theta^*(\omega)}\|^2\Big]
> > > > \end{equation}
> > > >
> > > > \begin{equation}
> > > > \text{s.t.} \quad \theta^*(\omega)=\arg \min_{\theta}\mathbb{E}_{ \mathcal{P}}\Big[\|R(s,a)+\gamma \psi(s')^\top \omega-\phi(s, a)^\top \theta\|^2\Big]
> > > > \end{equation}
> > > >
> > > > which has solution $(\theta_{\mathcal{P}},\omega_{\mathcal{P}})$ such that
> > > > $$
> > > > \theta_{\mathcal{P}}
> > > > = A_{\mathcal{P}}^\dagger b_{\mathcal{P},\omega_\mathcal{P}}
> > > > $$
> > > > Using a truncated SVD we obtain $\theta_r:=A_{\mathcal{P},r}^\dagger b_{\mathcal{P},\omega_\mathcal{P}}$ Therefore, the upper bound derived in the paper corresponds to the result presented in Equation (12).
> > > >
> > > > We have revised the paper to reflect these corrections.

---

> > > > ### Comment · Reviewer_ucxQ · 2025-11-26
> > > >
> > > > Thank all authors for answering my questions.
> > > >
> > > > I have a few follow-up questions.
> > > >
> > > > For the paper's positions, I still don't think AdaRL is a new framework from robust RL. Robust RL also has soft-target versions [1], which maximise the expectation over the dynamics distribution instead of the worst-case scenario. This is the same as the sampling dynamics in the paper. And the paper proposes to use an adaptive low-rank policy to better behave under this target.
> > > >
> > > > For the experimental designs, different physical parameters may have very different influences on the dynamics, and since the paper builds on the theoretical assumption of the Wasserstein distance, I think it should be proved at least on a toy example with such a property.
> > > >
> > > > [1] E. Derman, D. Mankowitz, T. Mann, and S. Mannor, “A Bayesian Approach to Robust Reinforcement Learning,” in Proceedings of The 35th Uncertainty in Artificial Intelligence Conference, PMLR, Aug. 2020, pp. 648–658. Accessed: Jan. 13, 2022. [Online]. Available: https://proceedings.mlr.press/v115/derman20a.html

---

> > > > > ### Author Response · Authors · 2025-11-26
> > > > >
> > > > > >For the paper's positions, I still don't think AdaRL is a new framework from robust RL. Robust RL also has soft-target versions [1], which maximise the expectation over the dynamics distribution instead of the worst-case scenario. This is the same as the sampling dynamics in the paper. And the paper proposes to use an adaptive low-rank policy to better behave under this target.
> > > > >
> > > > > **Response:** Thank you for the comment. We agree that some prior robust RL methods also consider expectations over a distribution of dynamics rather than a strict worst-case objective. That said, the key difference lies in the underlying mechanism.
> > > > >
> > > > > In Derman et al. (UAI 2020), robustness comes from averaging over a posterior distribution while keeping the policy class fixed. In our case, AdaRL adapts the effective rank of the policy/value representation in response to Wasserstein-bounded perturbations, controlling model complexity to improve stability under uncertainty. This adaptive low-rank adjustment is not present in those prior soft-target approaches.
> > > > >
> > > > > We appreciate the feedback and will tone down the use of the term “framework’’ in the revised paper to more clearly emphasize that our primary contribution is the adaptive low-rank mechanism rather than claiming an entirely new paradigm.
> > > > >
> > > > >
> > > > > >For the experimental designs, different physical parameters may have very different influences on the dynamics, and since the paper builds on the theoretical assumption of the Wasserstein distance, I think it should be proved at least on a toy example with such a property. we have tune down the usage of "framework" in our revision paper.
> > > > >
> > > > > **Response:**
> > > > > Thank you for the comment. As detailed in Appendix A.5.6, we provide a toy analysis based on the standard Gym CartPole environment to directly address this concern. In this example, we introduce uncertainty by varying the pole length across episodes, derive the corresponding linear dynamics, and compute a concrete Wasserstein bound $\varepsilon$ showing that the perturbed transitions remain within a valid Wasserstein ball.
> > > > >
> > > > > We further conduct a sanity-check experiment using models of different ranks. Although the nominal CartPole system has rank 4, the uncertain setting requires higher capacity, and the rank-8 model achieves the best performance—consistent with our theoretical prediction.
> > > > >
> > > > > Overall, this example demonstrates that physical parameter variations can be meaningfully captured under the Wasserstein formulation and that the theoretical behavior aligns well with empirical results.

---

### Official Review · Reviewer_niSV · 2025-10-31

**Soundness:** 3
**Presentation:** 3
**Contribution:** 3
**Rating:** 6
**Confidence:** 3

**Summary:**

The paper proposes AdaRL, a robust-RL method that improves generalization under epistemic dynamics uncertainty by adapting the rank of policy/critic representations during training. Uncertainty is modeled as a Wasserstein ball around a nominal kernel; trajectories are sampled from this ball rather than solving an inner worst-case problem. Learning is posed as a bi-level program: the lower level optimizes an entropy-regularized return under a fixed rank; the upper level selects the rank using a cumulative-spectrum rule and projects parameters to a low-rank manifold. Rank is updated at scheduled checkpoints. Experiments on MuJoCo (Hopper, Walker2d, Ant, Humanoid) show gains over fixed-rank SAC and robust baselines (RNAC, Parseval).

**Strengths:**

- Clear positioning versus min–max robust RL: AdaRL avoids repeatedly solving the inner worst-case kernel by sampling from a Wasserstein ball and controlling model capacity via rank. This is computationally attractive and easy to implement in deep RL.
- Principled and practical rank search: the cumulative singular-value rule plus projection yields a usable upper-level step; a hard-threshold alternative is ablated and shown to stagnate.
- Simple enforcement: the rank constraint is implemented by inserting a linear bottleneck between layers; occasional SVD refines factors.

**Weaknesses:**

- Ambiguity-set realism and certification: sampling uniformly from a Wasserstein ball is less conservative than worst-case evaluation. It would be valuable to report worst-case performance or couple AdaRL with a tractable DRO evaluation to support robustness claims.
- Baseline parity: RNAC is presented with its PPO implementation while AdaRL uses SAC; this can confound comparisons. Reference implementations exist and should be matched where possible.
- Exact placement of rank constraints: clarify which actor/critic layers are factorized, and add ablations (actor-only, critic-only, both).

**Questions:**

- Can you report performance under explicit worst-case dynamics for small tasks (e.g., via a tractable Wasserstein-DRO evaluation) to complement nominal and perturbed-physics tests?
- Do you adapt rank for both critics as well as the actor? Please include actor-only/critic-only/both ablations and a timing table for SVD/projection overhead.
- For RNAC, did you explore an SAC backbone or stronger PPO tuning to equalize compute and tuning budgets?

---

> ### Author Response · Authors · 2025-11-21
>
> > Ambiguity-set realism and certification: sampling uniformly from a Wasserstein ball is less conservative than worst-case evaluation. It would be valuable to report worst-case performance or couple AdaRL with a tractable DRO evaluation to support robustness claims.
>
> **Response**: We thank the reviewer for this insightful question. The key design objective of our method is precisely to avoid computing the worst-case transition kernel required by DRO-based robust RL. Existing RMPD formulations [1] are tractable only in finite-state, finite-action settings with carefully constructed convex ambiguity sets—assumptions that are far from the high-dimensional continuous-control environments we study. Moreover, it is well known that robust MDPs with general (non-rectangular) ambiguity sets are NP-hard or intractable, and tractability requires strong structural assumptions [2].
>
> For this reason, our goal is not to approximate the inner maximization of a classical DRO formulation. Instead, AdaRL intentionally bypasses the nested min–max structure by leveraging the observation that overly conservative worst-case dynamics often lead to unnecessary rank inflation and degraded performance. The sampling-based inner loop serves only to estimate the average-case Bellman error under plausible perturbations, while the outer loop adaptively adjusts the policy rank to match the intrinsic dimensionality of the task.
>
> Thus, AdaRL is not intended to be a strict DRO solver; rather, it provides a computationally efficient and substantially less conservative alternative for achieving robustness in continuous-control problems.
>
>
> Reference: [1] Xu, Huan, and Shie Mannor. "Distributionally robust Markov decision processes." Advances in Neural Information Processing Systems 23 (2010).
> [2] Wiesemann, Wolfram, Daniel Kuhn, and Berç Rustem. "Robust Markov decision processes." Mathematics of Operations Research 38.1 (2013): 153-183.
>
> > Exact placement of rank constraints: clarify which actor/critic layers are factorized, and add ablations (actor-only, critic-only, both).
>
> **Response**: We thank the reviewer for pointing this out. We have added the requested ablations—actor-only, critic-only, and both—in the revised submission. The exact placement of the rank constraints and the corresponding experimental results are now provided in Appendix A.5.5.

---

> > ### Author Response · Authors · 2025-11-21
> >
> > > Q1) Can you report performance under explicit worst-case dynamics for small tasks (e.g., via a tractable Wasserstein-DRO evaluation) to complement nominal and perturbed-physics tests?
> >
> > **Response**: Thank you for the question. As discussed in our previous response, computing explicit worst-case dynamics under a Wasserstein uncertainty set is not feasible in continuous-control environments. Identifying an adversarial kernel requires solving an infinite-dimensional functional optimization problem that cannot be implemented within this simulators.
> >
> > To address your concern, we have expanded our experiments to explicitly compare AdaRL and the strongest baselines under worst-case outcomes obtained through uniform sampling from the uncertainty set. In Appendix Figure 8, we present the worst-case performance for each method and further increase the number of evaluation rollouts per dynamic to ensure fairness. The results show that in the vast majority of cases, AdaRL achieves a better worst-case score than the baselines; and even in the few cases where AdaRL’s worst-case performance is slightly lower, its average performance remains substantially higher.
> >
> > These new results have been included in the revised manuscript and further demonstrate the robustness advantages of the adaptive-rank mechanism.
> >
> > > Q2) Do you adapt rank for both critics as well as the actor? Please include actor-only/critic-only/both ablations and a timing table for SVD/projection overhead.
> >
> > **Response**: Thank you for the question. In our implementation, we apply rank adaptation only to the policy network. Following your suggestion, we have added the requested ablations in the revised manuscript. Appendix A.5.5 now reports results for actor-only, critic-only, and actor–critic rank adaptation. As summarized in Table 2, the critic-only variant yields only limited robustness gains, while actor-only already provides strong improvements. The actor–critic configuration consistently achieves the best overall performance. These results clarify how applying rank adaptation to different components influences robustness.
> >
> >
> > > Baseline parity: RNAC is presented with its PPO implementation while AdaRL uses SAC; this can confound comparisons. Reference implementations exist and should be matched where possible.
> >
> > > Q3) For RNAC, did you explore an SAC backbone or stronger PPO tuning to equalize compute and tuning budgets?
> >
> > **Response**: We thank the reviewer for the helpful suggestion. During the rebuttal phase, we conducted two additional investigations to strengthen the fairness and completeness of our comparison.
> >
> > First, within the RNAC framework, we evaluated both variants provided in the official repository—the DS-based method (RNAC-DS) and the IPM-based method (RNAC-IPM). Across multiple runs, we found that RNAC-IPM consistently produces more robust policies than RNAC-DS in our setting. We believe this improvement stems from the fact that the sampling-based worst-case value estimation used in DS can introduce significant bias under rapidly varying dynamics, whereas the IPM formulation—implemented through a regularization-based approach—provides a more stable and conservative surrogate. We have updated the corresponding table in the manuscript to include these results.
> >
> > Second, we also explored integrating an SAC backbone to more closely match compute and tuning budgets. However, these efforts were limited by the legacy software stack used in the official RNAC implementation (MuJoCo 2.1.2.14 and Gym 0.21.0). In this setup, each rollout requires switching environments, which substantially slows down sampling and renders extensive tuning prohibitively time-consuming (e.g., nearly one day per seed). Within the feasible number of runs, we tested several SAC configurations but did not observe performance improvements over the reported RNAC results.

---

> > > ### Comment · Reviewer_niSV · 2025-11-25
> > >
> > > Thank you for your detailed response. Your clarifications were helpful, and I will keep my original score.

---

> > > > ### Author Response · Authors · 2025-11-26
> > > >
> > > > Thank you for your thoughtful review and for taking the time to consider our clarifications. We appreciate your feedback and are grateful for your recognition.

---

### Official Review · Reviewer_5kUX · 2025-11-01

**Soundness:** 2
**Presentation:** 3
**Contribution:** 2
**Rating:** 2
**Confidence:** 3

**Summary:**

This paper aims to tackle the problem of epistemic uncertainty in robust reinforcement learning (RL) by providing a bi-level optimization framework that analyzes the bias-variance trade-off in entropy regularized RL. This framework combats the over-conservatism of a learned policy that often arises due to the nested min-max structure of the optimization problem. This is accomplished by optimizing under fixed, low-rank constraints when in the inner loop, then dynamically adjusting the rank in the outer loop and projecting the parameters on to a low-rank manifold.

**Strengths:**

1. The paper was neat, organized, and well written. It naturally follows and addresses some of the questions that would arise as a reader is going through it. The authors did a good job discussing the bias-variance tradeoff.
2. Adaptively controlling the policy's rank was a fascinating twist, allowing for the incorporation of conservatism during learning. Namely, I found the improved efficiency particularly interesting as it stemmed from repeatedly swapping between traditional policy optimization with a fixed rank to dynamically change the rank.
3. I found the way the algorithm was structured interesting. Namely, by swapping between the lower-level policy optimization and upper level adaptation of the rank and subsequent projection.
4. The experiments were conducted on varied environments, showing empirical validation for the author's claims. Additionally, the author backs up their claims with recent relevant work [1].

[1] Saket Tiwari, Omer Gottesman, and George Konidaris. Geometry of neural reinforcement learning in continuous state and action spaces. 2025.

**Weaknesses:**

1. While I found the problem that this paper tackles interesting, the work surrounds a single theorem with minimal additional contributions while relying on some rather strong assumptions. Ultimately, this may not sufficiently capture the complex nuances of more complicated tasks.
2. There is a gap between the theoretical analysis and the practical implementation of the experiments. Namely, the theory analyzes rank $r$ of the covariance matrix in a linear model, but in Appendix A.4.1, the authors the authors instead constrain the rank of the weight matrix $W$ within a non-linear deep neural network. The authors perform a sanity check in figure 2, and while doing so is responsible, it is not a substitute for a formal theoretical argument.
3. The authors briefly discuss the difficulty of deriving convergence guarantees, however, such results are not presented in this work. Though an interesting problem to tackle, the only theoretical result being Theorem 1 leaves significant open questions in this topic.
4. On line 417, it is mentioned that environment dynamics vary across episodes. How are these dynamics varied?
5. Is the uncertainty set also compact?

**Questions:**

1. Line 112, specifying a "Discounted Markov decision process" helps provide additional clarity between this setting and the average reward setting.
2. Typo on line 142. It looks like there is a period instead of a comma.
3. It would be nice to see $\pi^*_\mathcal{P}$ defined precisely where it is first mentioned on line 148.
4. I would've liked to see all three assumptions listed in the main body prior to theorem 1.
5. Repeated/similar wording on lines 243 and 246.
6. Extra period on line 312.
7. $\tau$ is not defined in equation 12.
8. Line 740 mislabels the assumption.

---

> ### Author Response · Authors · 2025-11-21
>
> > 1. While I found the problem that this paper tackles interesting, the work surrounds a single theorem with minimal additional contributions while relying on some rather strong assumptions. Ultimately, this may not sufficiently capture the complex nuances of more complex environments.
>
> **Response**: We thank the reviewer for this thoughtful comment. Regarding the assumptions, Assumptions 1 and 2 are standard in the RL literature and are routinely used to obtain value-function approximation guarantees. We agree that Assumption 3 is stronger, though related spectral-decay assumptions have appeared in prior RL analyses.
>
> In response to the reviewer’s concerns, we have substantially revised and expanded the theoretical section in the appendix. Concretely:
>
> 1. We restate the main bias–variance result and show that the fundamental robustness–rank tradeoff continues to hold under standard assumptions, without requiring the Discrete Picard condition.
> 2. We extend the analysis to nonlinear function classes by deriving an analogous bias–variance decomposition in the infinite-width (Neural Tangent Kernel) regime, demonstrating that the rank tradeoff is not limited to linear models.
> 3. We provide an additional version of the result that avoids Assumption 3 entirely, yielding a more assumption-light justification for the observed behavior.
>
> These revisions clarify that our conclusions are not tied to overly restrictive assumptions and that the core robustness–low-rank mechanism persists across both linear and nonlinear settings.
>
> Finally, we respectfully disagree with the reviewer’s statement that our framework “does not sufficiently capture the complex nuances of more complex environments.”
> Even in high-dimensional environments such as Humanoid-v3 (whose observation space has 376 dimensions), a large fraction of the dimensions correspond to joint angles, orientations, and kinematic quantities. Although the observation dimension is large, the effective degrees of freedom governing the dynamics remain much lower due to physical constraints and Newtonian mechanics. This implies that both the value function and the optimal policy naturally exhibit a low intrinsic rank, making them prone to overparameterization effects when represented with large neural networks. Our empirical results on MuJoCo benchmarks corroborate this observation and illustrate that adaptive rank selection provides meaningful robustness benefits precisely in such complex environments.
>
>
> > 2. There is a gap between the theoretical analysis and the practical implementation of the experiments. Namely, the theory analyzes rank r of the covariance matrix in a linear model, but in Appendix A.4.1, the authors here instead constrain the rank of the weight matrix W within a non-linear deep neural network. The authors perform a sanity check in Figure 2, and while doing so is responsible, it is not a substitute for a formal theoretical argument.
>
> **Response**: Thank you for the comment. We agree that the original version focused on the linear-feature setting, while the experiments use nonlinear neural networks. To bridge this gap, the revised manuscript now includes an explicit analysis of the nonlinear case. In particular, Appendix A.3-A.4 has been extended to provide a bias–variance decomposition in the infinite-width (Neural Tangent Kernel) regime, showing that the same robustness–rank tradeoff holds for nonlinear networks. This establishes a direct theoretical connection to the practical implementation, where the rank constraint is imposed on deep network layers.
>
> These additions ensure that our theoretical framework aligns with the nonlinear architectures used in the experiments and clarify why the rank-adaptation mechanism behaves consistently across both settings.
>
> > 3. The authors briefly discuss the difficulty of deriving convergence guarantees, however, such results are not presented in this work. Though an interesting problem to tackle, the only theoretical result being Theorem 1 leaves significant open questions in this topic.
>
> **Response**: We appreciate the reviewer’s question. A complete convergence theorem for entropy-regularized actor–critic methods under model uncertainty, deep parametrization, and bi-level rank adaptation is indeed extremely challenging and beyond the scope of this work. However, we have expanded the discussion in Appendix A.4 to clarify the heuristic convergence mechanism. In particular, we outline how the geometry of low-rank matrix varieties and the design of our rank-update rule together provide a coherent rationale for the stability and convergence behavior observed in our experiments.

---

> > ### Author Response · Authors · 2025-11-21
> >
> > > 4. On line 417, it is mentioned that environment dynamics vary across episodes. How are these dynamics varied?
> >
> > **Response**: The details of the experimental setup are provided in Appendix A.5.2, along with an illustrative example in Figure 6. Briefly, the environment dynamics are varied by introducing structural and physical perturbations. For Hopper, we modify agent morphology (e.g., torso and foot sizes). For Ant and Humanoid, we alter physical parameters such as gravity and apply external forces (e.g., wind with a specified velocity).
> >
> > > 5. Is the uncertainty set also compact?
> >
> > **Response**: Yes. It is a closed ball in Wasserstein metric.
> >
> >
> > > **Q1)** Line 112, specifying a "Discounted Markov decision process" helps provide additional clarity between this setting and the average reward setting.
> >
> > **Response**: Thank you for the suggestion. We have updated the text to explicitly specify the discounted Markov decision process setting for greater clarity.
> >
> >
> > > **Q2)** Typo on line 142. It looks like there is a period instead of a comma.
> >
> > **Response**: We appreciate the reviewer for catching this typo. The period on line 142 has been corrected to a comma in the revised version.
> >
> >
> > > **Q3)** It would be nice to see $\pi_P^*$ defined precisely where it is first mentioned on line 148.
> >
> > **Response**: Thank you for pointing this out. We have updated the manuscript to provide a precise definition of $\pi^{*}_{\mathcal{P}}$. Here $\pi^{*}_{\mathcal{P}}$ denotes the optimal policy under the robust RL formulation.
> >
> >
> > > **Q4)** I would've liked to see all three assumptions listed in the main body prior to theorem 1.
> >
> > **Response**: Thank you for the suggestion. We have now moved all three assumptions to the main body before Theorem 1 for improved clarity.
> >
> >
> > > **Q5)** Repeated/similar wording on lines 243 and 246.
> >
> > **Response**: Thank you for pointing this out. We have revised the text to remove the redundant wording and improved the clarity of the paragraph. The updated version now avoids repetition while preserving the original meaning.
> >
> >
> > > **Q6)** Extra period on line 312.
> >
> > **Response**: We appreciate the careful proofreading. The extra period on line 312 has been corrected.
> >
> >
> > > **Q7)** $r$ is not defined in equation 12.
> >
> > **Response**: Thank you for pointing this out. We have clarified the definition of $r$ in the revised manuscript. $r$ denotes the rank variable.
> >
> >
> >
> > > **Q8)** Line 740 mislabels the assumption.
> >
> > **Response**: Thanks for the suggestion. The assumption is "Assume $\mathcal{P}^{\circ} \in \mathcal{B}_W(\hat{\mathcal{P}}^\circ,\epsilon)$". We have corrected it in the revised version.

---

> > > ### Author Response · Authors · 2025-11-26
> > >
> > > > 2. There is a gap between the theoretical analysis and the practical implementation of the experiments. Namely, the theory analyzes rank r of the covariance matrix in a linear model, but in Appendix A.4.1, the authors here instead constrain the rank of the weight matrix W within a non-linear deep neural network. The authors perform a sanity check in Figure 2, and while doing so is responsible, it is not a substitute for a formal theoretical argument.
> > >
> > >
> > > **Response**: Thank you for the comment. To better connect the linear theory with practice, we added a concrete toy example in Appendix A.5.6 using the Gym CartPole environment. By varying the pole length, we derive the corresponding linear dynamics and compute an explicit Wasserstein bound $\varepsilon$, showing that the perturbed transitions remain within a valid Wasserstein ball.
> > >
> > > We also run a simple rank comparison, where the uncertain setting favors rank 8—consistent with our theoretical prediction. We hope this concrete example helps address your concern about the theory–experiment connection, and we look forward to any further feedback you may have.

---

### Official Review · Reviewer_8nqM · 2025-11-01

**Soundness:** 2
**Presentation:** 3
**Contribution:** 2
**Rating:** 4
**Confidence:** 3

**Summary:**

The guiding principal of the paper is 'simple models (non-overly fitted) generalizes better'.  The paper tries to learn low rank policy for the given nominal environment which may perform better for other environment in the uncertainty set. It adaptively learn the rank of the policy.

**Strengths:**

The idea of aproaching robust RL from low-rank perspective is potent.

**Weaknesses:**

See questions below.

**Questions:**

Q1) The approach remains the same for the all uncertainty levels. That is, suppose we are given two secnarios: First where model uncertainty is small and other where its large. The two scenarios  have different robust optimal policies, however the approach in the paper remains the same for two, hence rendering the same policy. How it can be tackled effectively.

Q2) The intuitive relation between low rank policies and robustness is interesitng, however I don't see sufficient concrete theoretical support for it.

Q3) How can this approach be extended to large MDPs in online settings?

---

> ### Author Response · Authors · 2025-11-21
>
> > Q1) The approach remains the same for the all uncertainty levels. That is, suppose we are given two scenarios: First where model uncertainty is small and other where its large. The two scenarios have different robust optimal policies, however the approach in the paper remains the same for two, hence rendering the same policy. How it can be tackled effectively.
>
> **Response:**  Thank you for the thoughtful question. We would like to clarify that although the algorithmic procedure is identical across different uncertainty levels, the resulting policies are not. This is because the rank-adaptation mechanism is explicitly driven by the magnitude of epistemic uncertainty. As the uncertainty radius increases, the selected ranks change accordingly, leading to different effective model classes and, ultimately, different robust policies.
>
> To further illustrate this effect, we have added an ablation study in Appendix A.5.5 in the revised version. The results in Table 2 show that the learned policies differ across low- and high-uncertainty settings. These findings confirm that AdaRL adapts its policy behavior in response to the level of model uncertainty.
>
>
>
> > Q2) The intuitive relation between low rank policies and robustness is interesting, however I don't see sufficient concrete theoretical support for it.
>
>
> **Response:** We thank the reviewer for raising this point. We acknowledge that in the original submission, our theoretical analysis was first presented in the setting of linear parameterizations, which may have given the impression that the robustness–low-rank connection is specific to that case. In the revised version, we clarify that the underlying mechanism is not restricted to linear models. For general function classes, any finite-width network spans only a finite-dimensional manifold of Q-functions. This induces approximation bias, while Wasserstein perturbations introduce a form of variance reduction along directions aligned with that manifold mirroring the same bias–variance tradeoff characterized in the linear setting.
>
> To make this connection precise, the revised manuscript now includes a bias–variance decomposition in the arbitrarily wide (NTK)[1] regime (Appendix A.3–A.4). This analysis formalizes how low-rank structure in the value-function class interacts with distributional perturbations, thereby providing more rigorous theoretical support for the robustness benefits of low-rank policies.
>
> Reference:[1]Jacot, Arthur, Franck Gabriel, and Clément Hongler. "Neural tangent kernel: Convergence and generalization in neural networks." Advances in neural information processing systems 31 (2018).
>
>
> > Q3) How can this approach be extended to large MDPs in online settings?
>
> **Response:**  We thank the reviewer for this thoughtful question. We would first like to kindly clarify that the *algorithmic* component of AdaRL is inherently designed for online RL. In our experiments, we have already tested the method on high-dimensional MuJoCo environments (e.g., Ant with a 105-dimensional observation space and Humanoid-v3 with 348 dimensions), which empirically demonstrates that the approach remains stable and effective in large, continuous MDPs.
>
> For even larger MDPs, one might worry about issues such as higher state dimensionality, increased variance in online sampling, or the computational cost of adaptation steps. We believe these concerns remain manageable: AdaRL relies only on local online sampling, lightweight rank updates, and a low-rank structure that reduces the effective parameter dimension. Online sampling integrates naturally into the bi-level actor–critic framework; rank adaptation requires only a truncated SVD of a single weight matrix and is performed infrequently; and the resulting low-rank parameterization serves as a structural regularizer that improves sample efficiency and maintains scalability even in large, high-dimensional MDPs.

---

### Author Response · Authors · 2025-11-21
**Rebuttal Summary**

Dear reviewers and AC,

We thank all reviewers for their valuable comments. In response, we have revised the original submission. The main updates are summarized below:

1. **New theoretical support** (Reviewer `5kUX`):
   In Appendix A.3, we extend our analysis to nonlinear function classes by deriving an analogous bias–variance decomposition in the infinite-width (Neural Tangent Kernel) regime, showing that the rank trade-off is not limited to linear models. We also provide an alternative version of the result that removes Assumption 3 entirely, offering a more assumption-light justification for the observed behavior. Appendix A.4 further includes a geometric interpretation of the proposed framework.

2. **Expanded baseline comparisons and ablation studies** (Reviewers `niSV` and `8nqM`):
   We added additional RNAC baselines and new ablation studies. The corresponding results are reported in Section 5 and in Appendices A.5.4 and A.5.5.

3. **Broader discussion of Theorem 1** (Reviewer `5kUX` and `ucxQ`):
   We expanded the theoretical discussion in **Section 3** to present results both with and without Assumption 3. We also refined the notation following the suggestions from Reviewers `ucxQ` and `5kUX`.



Best regards,
Authors of Submission 9826

---

### Author Response · Authors · 2025-11-30
**Overview of Key Changes and Improvements**

Dear Area Chair and Reviewers,

Thank you for handling our submission. Following the discussions with the reviewers, we have substantially revised both the theoretical and empirical parts of the paper. Below we summarize the key updates and how they directly address the concerns raised by the reviewers.

---

### **1. Motivation & Framework Clarification (Reviewers `niSV`, `8nqM`)**

Following the reviewers’ comments, we clarified and re-emphasized our motivation for avoiding the inner worst-case optimization in robust RL (revised Figure 1). Traditional robust RL methods depend on solving a nested min–max or DRO-style worst-case problem, which is computationally expensive and often overly conservative in high-dimensional settings.

In the revision, we make explicit that AdaRL is designed to avoid this bottleneck: instead of optimizing the inner DRO problem, we sample dynamics from a Wasserstein ball and adapt the policy rank accordingly, yielding a practical, tractable, and effectively robust alternative to classical min–max formulations.

---

### **2. Expanded Theory: Removing Strong Assumptions & Adding the Nonlinear Case (Reviewers `5kUX`, `ucXQ`)**

Reviewer `5kUX` raised two key concerns:
(1) our theoretical analysis relied on an overly strong **Assumption 3 (Discrete Picard Condition)**, and
(2) the theory only covered the **linear** setting, creating a gap with nonlinear neural networks used in experiments.

To address these concerns:

- **We provide two versions of the main result**:
  - one version **without** Assumption 3, relying only on standard RL approximation assumptions;
  - one version **with** Assumption 3, providing a sharper but more restrictive statement.

- **We added a full nonlinear extension**, deriving an analogous bias–variance decomposition in the NTK/infinite-width regime (Appendix A.3–A.4). This demonstrates that the robustness–rank mechanism extends naturally to nonlinear function classes.

- **We added concrete examples for both linear and nonlinear cases**:
  - A *linear CartPole example* (Appendix A.5.6) where we vary the pole length, derive the perturbed linear dynamics, and compute an explicit Wasserstein radius ε.
  - A *nonlinear toy example* in the appendix to illustrate the NTK-based results.

Across these examples, the empirical behavior aligns closely with our theory, strengthening the justification for our adaptive-rank framework.

---

### **3. Enhanced Experiments, Baselines, and Ablation Studies (Reviewers `niSV`, `8nqM`)**

Based on the feedback from Reviewers `niSV` and `8nqM` we significantly expanded our experimental section:

- **Stronger and more complete baselines**
  We added additional RNAC baselines and clarified backbone choices to ensure fairness in comparisons with AdaRL.

- **New Ablation Studies**
  Following the reviewers’ suggestions, we added ablations on:
  - actor-only rank adaptation,
  - critic-only adaptation,
  - joint rank adaptation,
  These new results (Section 5 and Appendix A.5.5) substantially strengthen the empirical validation.

Overall, these updates ensure a more comprehensive and transparent comparison, addressing their concerns about missing baselines and limited ablations.

---

### **Overall**

We have carefully addressed all reviewer concerns through substantial updates in motivation, theory, and experiments.
**The revised version now provides clearer motivation, more general and assumption-light theoretical guarantees, and a richer experimental evaluation**, and we believe all major issues raised during the review process have been resolved.

Thank you for your time and consideration.

Best regards,

Authors

---

### Note · Program_Chairs · 2026-01-17
**Submission Desk Rejected by Program Chairs**

The following references in this submission do not refer to real documents and/or have major errors in bibliographic information:

 Zichuan Yang, George Tucker, Tom Zahavy, Mohammad Ghavamzadeh, and Ofir Nachum. Representation learning for reinforcement learning via bellman error minimization. In International Conference on Machine Learning (ICML), 2020. URL https://arxiv.org/abs/2001.07301.